# Chronic pain: Evidence from the national child development study

**David G. Blanchflower[1], Alex Bryson [2]***

**1** Adam Smith School of Business, Dartmouth College, University of Glasgow and NBER, Hanover, NH, United States of America, **2** UCL Social Research Institute, University College London, London, United Kingdom

* a.bryson@ucl.ac.uk

**Data Availability Statement:** All sweeps of the NCDS are available to academics in the ESRC Data Archive. See the weblink here: https://beta. ukdataservice.ac.uk/datacatalogue/studies/? Search=National+Child+Development+Survey#!? Search=National%20Child%20Development%

## Abstract

Using data from all those born in a single week in 1958 in Britain we track associations between short pain and chronic pain in mid-life (age 44) and subsequent health, wellbeing and labor market outcomes in later life. We focus on data taken at age 50 in 2008, when the Great Recession hit and then five years later at age 55 in 2013 and again at age 62 in 2021 during the Covid pandemic. We find those suffering both short-term and chronic pain at age 44 continue to report pain and poor general health in their 50s and 60s. However, the associations are much stronger for those with chronic pain. Furthermore, chronic pain at age 44 is associated with a range of poor mental health outcomes, pessimism about the future and joblessness at age 55 whereas short-duration pain at age 44 is not. Pain has strong predictive power for pain later in life: pain in childhood predicts pain in mid-life, even when one controls for pain in early adulthood. Pain appears to reflect other vulnerabilities as we find that chronic pain at age 44 predicts whether or not a respondent has Covid nearly twenty years later.

## 1. Introduction

*"To live with chronic pain is to live with daily challenges around simple tasks that others take for granted. It often means being disbelieved, stigmatised for not getting better, or judged as not coping. It might mean living with poor mental health and self-esteem, absenteeism from school or work, the breakdown of relationships, and socioeconomic disadvantage. For society, the costs are staggering: low back pain is the leading cause of years lost to disability and chronic pain costs billions of dollars through health system expenditures, productivity losses, reduced quality of life, and informal care. . . . . .Chronic pain is real. It deserves to be taken more seriously."*

The Lancet editorial May 29[th], 2021

Advances are being made in medicine in understanding the nature of chronic pain and how to treat it, as the special issue of *The Lancet* devoted to the issue indicates. There is also an

20Survey&Page=1&Rows=10&Sort=
1&DateFrom=440&DateTo=2022 Researchers can
find further guidance and assistance from the
Centre for Longitudinal Studies here: https://cls.
ucl.ac.uk/cls-studies/1958-national-child-
development-study/ We received no special
privileges in accessing the data.

**Funding:** Alex Bryson thanks the Health Foundation
for funding (grant number 789112). The funders
had no role in study design, data collection and
analysis, decision to publish, or preparation of the
manuscript.

**Competing interests:** The authors have declared
that no competing interests exist.

appreciation of the problems it brings to suffers in daily life. Chronic pain is a very serious problem affecting a large number of people. Krueger [1] found that nearly half of prime age men in the United States who are not in the labor force take pain medication on any given day; and in nearly two-thirds of these cases, they take prescription pain medication. According to the National Academies of Sciences, Engineering and Medicine [2] more than 100 million Americans suffer from chronic pain, that is, pain lasting at least three months.

We contribute to the literature by investigating links between pain experienced across the life-course and subsequent health, wellbeing and labor market outcomes among those born in Britain in a single week in 1958. Most longitudinal investigations of links between pain and subsequent health and labor market outcomes focus on short-term change captured over the space of a year, sometimes two, and are thus unable to shed any light on the longer-term effects of pain. Our data permit us to examine these longer-term relationships over decades, beginning with experience of pain in childhood, to assess the effects of pain on a wider range of health, wellbeing and labor force outcomes than is ordinarily available in a single data set. We show that those reporting aches and pains at age 44 report poorer health, wellbeing and labor market outcomes over the following two decades through age 62. The partial associations between chronic pain and subsequent health and labor market outcomes persist even in the presence of lagged pain measured earlier in life and having controlled for parental and familial background in childhood, as well as a wide range of physical and mental health ailments reported in mid-life.

We also find that chronic pain at age 44 in 2002 predicts having Covid nearly two decades later in 2021. This suggest that pain is picking up broader characteristics such as general health, sensitivity, mental health etc. which seem to make the respondent vulnerable to catching Covid.

The remainder of the paper is structured as follows. Section Two reviews the literature on associations between pain, health, wellbeing and labor market outcomes. Section Three provides some context for our analysis by reporting the incidence of pain in the UK relative to other countries and identifying some of its correlates. Section Four introduces the National Child Development Study (NCDS) data. Section Five reports results before we conclude in Section Six with a reflection on the implications of our findings.

## 2. Literature review

In the short-term pain can be distressing and debilitating, but it may be particularly problematic if it persists into the longer-term ("chronic" pain), and if it has consequences in the future for people's physical or mental health, their wellbeing, their labor market participation or prospects for a good quality of life. Here we review the association between pain and subsequent health, labor market participation and well-being in turn.

### 2.1. Pain and subsequent health

A number of studies investigate predictors of chronic widespread pain in adulthood, including pain experienced in childhood. Some examine what is sometimes termed the "early pain pathway" by which common childhood symptoms like abdominal pain and headaches or migraine may be linked to subsequent pain. The empirical evidence is mixed. Using data from NCDS Jones et al. [3] find evidence that multiple common symptoms (including abdominal pain and migraine/headaches) in childhood at ages 7, 11 and 16 are associated with chronic widespread pain by age 45. The size of the effect was not large, and only relatively few children experienced multiple symptoms, so the associated population attributable risk was low.

A study using an earlier British population-based birth cohort study established in 1946 assessed the association between abdominal pain in childhood and subsequent physical and

psychiatric health in adulthood [4]. It found children with persistent abdominal pain were more likely to suffer psychiatric disorders in adulthood than those who did not suffer from childhood abdominal pain, but they were not especially prone to physical problems once psychiatric disorders were controlled for. Using the National Longitudinal Study of Adolescent to Adult Health (Add Health) Noel et al. [5] also show that those experiencing chronic pain in adolescence subsequently report higher rates of lifetime anxiety and depressive disorders compared to individuals without a history of adolescent chronic pain.

There is compelling evidence regarding the persistence of chronic pain in adulthood provided by Mundal et al [6] in their study of a large Norwegian prospective general population cohort study. They find chronic widespread pain persists 11 years later among half of those who initially reported it with poor sleep, obesity and chronic disease all predicting its persistence. Similarly, in Croft et al.'s [7] study of consultation data from general practice records for over 10,000 women the strongest predictor of current pain was episodes of musculoskeletal illness and mental disorders 15 to 25 years earlier. The strongest predictor of head and neck pain was earlier migraine, while back pain was most strongly associated with earlier back complaints, abdominal pain and earlier intestinal-related problems. However, chest pain and the presence of widespread pain were more strongly associated with previous mental health problems, rather than with region-specific illnesses.

Other cohort studies focus on other predictors of chronic pain in adulthood. For instance, Pang et al. [8] show behavioural problems in childhood among the 1958 NCDS cohort predict chronic widespread pain in adulthood. Similarly, Jones et al. [9] examine the childhood antecedents to chronic widespread pain at age 45 in the NCDS. Using mothers' reports of their children's experiences by age 7 they find certain traumatic events, such as being hospitalised following a road traffic accident, do predict chronic widespread pain in adulthood. Jay et al. [10] show that hardship in early adulthood is associated with increased risk of chronic widespread pain by age 68 in the 1946 British birth cohort study.

Fine [11] argued that chronic pain "*negatively impacts multiple aspects of patient health, including sleep, cognitive processes and brain function, mood/mental health, cardiovascular health, sexual function, and overall quality of life. Furthermore, chronic pain has the capacity to become increasingly complex in its pathophysiology, and thus potentially more difficult to treat over time. The various health complications related to chronic pain can also incur significant economic consequences for patients*."

Fewer studies consider the consequences of pain in mid-life for subsequent health outcomes. One of the few that does finds women reporting severe bodily pain suffer cognitive impairment 13 years later but pain did not significantly affect mortality rates [12]. However, a study of 6,940 individuals recruited from 29 practices across Grampian in the North-East of Scotland found severe chronic pain is associated with all-cause mortality ten years later, and particularly circulatory system disease-related death [13].

There are some studies that question the importance of chronic pain for subsequent health outcomes. For example, in their study of 1,953 individuals, McBeth et al. [14] find that whilst those with chronic widespread pain at baseline experienced increased levels of psychological distress at follow up 12 months later, the association was accounted for by concomitant features of chronic pain (the presence of other physical and psychosocial factors) rather than pain per se.

## 2.2. Pain and subsequent labor market participation

It is well-known that pain is associated with unemployment and economic inactivity [1]. Blanchflower and Bryson [15] show the unemployed suffer greater pain than the employed

across the life-course, with pain incidence being highest among those not in the labor force (NILF) from age 30 onwards. A study using the NCDS used in this paper finds those who are unemployed at age 45 are more likely to report chronic widespread pain [16], while Macfarlane et al. [17] show musculoskeletal pain at age 45 is greater among those in lower social classes. A similar cross-sectional correlation between pain and joblessness is found in other studies [18–20].

Recently, these studies of cross-sectional associations have been supplemented with studies using panel data which show pain leads to joblessness and that the effects are larger among those who experience frequent pain. For instance, using data from the German Socio-economic Panel (GSOEP) for the period 2002 to 2018 Piper et al [21] find those reporting pain 'always' are 4–5 percentage points more likely than those suffering no pain to be jobless a year later, were more likely to have cut their working hours if still in work, and to work below their desired number of hours. Effects were robust to controlling for life satisfaction, mental health, and employee's occupation at baseline, and persisted with the inclusion of person fixed effects accounting for unobserved fixed differences according to the reporting of pain and were apparent two years later.

Similarly, Virtanen et al. [22] find musculoskeletal pain is associated with subsequent prolonged unemployment between age 31–42 among a small cohort of individuals in Northern Sweden. A study of patients attending pain clinics in the Pacific Northwest of the United States shows a variety of pain measures predict unemployment and being unable to work six months later [23], while Kaspersen et al. [24] find those in the Norwegian HUNT study with multiple musculoskeletal pain symptoms spend more time in unemployment over the subsequent 14 years, even conditioning on a wide range of other physical and psychological health problems.

## 2.3. Pain and subsequent well-being

It is well-known that happiness and subjective well-being more broadly (including life satisfaction, optimism and positive emotions) lead to better health and longevity [25, 26]. However, the relationship may be bi-directional, with chronic pain and poor health leading to changes in subjective well-being [27]. Reviewing the literature Eccleston [28 pp. 99] argues that "chronic pain can detrimentally alter natural ageing and halt personal development. Critical to this is the subjective judgement of who one is". He draws attention to qualitative research indicating that pain challenges individuals' core sense of who one is, distorting identity: "*one's identity can be enmeshed so closely with pain that it becomes difficult to think of any aspect of one's life as being free of pain, including one's idea about any future*" (p. 94). Sturgeon et al. [29] identify pain as catastrophizing, and the perceived injustice in suffering pain as factors contributing to pain-related distress and increased risk of long-term disability through maladaptive behavioral responses. IsHak et al [30] have noted that "*pain and depression are highly intertwined and may co-exacerbate physical and psychological symptoms. These symptoms could lead to poor physical functional outcomes and longer duration of symptoms.*"

Perhaps the most robust evidence of a link between pain and subsequent subjective well-being is the paper by McNamee and Mendolia [31]. Using panel data from the first ten waves of the Household, Income and Labour Dynamics of Australia Survey (HILDA) they show chronic pain leads to lower life satisfaction, a finding that is robust to the inclusion of person fixed effects. They find some adaptation over a three-year period, though the effects of chronic pain persist.

A related literature examines the relationship between pain and sleep. A number of cross-sectional studies point to an association between pain and sleep problems. It is often assumed that the association is bi-directional with pain having a sleep-interfering property and sleep

deprivation having a pain-enhancing effect. However, this has recently been challenged. In their daily process study Tang et al. [32] use an electronic diary to record patients' self-reported sleep quality and efficiency, together with ratings of pain, mood and arousal at different times of the day, with sleep efficiency captured using actigraphy. They find pre-sleep pain was not a reliable predictor of subsequent sleep, whereas sleep quality was a consistent predictor of pain the next day.

In contrast, Nicassio et al [33] found that, among a sample of patients diagnosed with rheumatoid arthritis prior pain predicted subsequent sleep problems, whereas sleep problems did not affect subsequent pain. There is also evidence that among individuals with acute low back pain poor sleep quality was associated with subsequent pain intensity [34]. The effect was large and robust to controlling for symptoms of depression. A recent review of the literature concludes that insomnia symptoms predispose individuals to chronic pain or the worsening of painful conditions, but "*there are still few longitudinal studies investigating sleep disturbances as a possible pathogenic condition of chronic pain*" [35, p. 1249].

Most longitudinal investigations of links between pain and subsequent health and labor market outcomes focus on short-term change captured over the space of a year, sometimes two, and are thus unable to shed any light on the longer-term effects of pain. We contribute to the literature with data permitting us to examine longer-term relationships over decades, beginning with experience of pain in childhood, to assess the effects of pain on a wider range of health, wellbeing and labor force outcomes than is ordinarily available in a single data set. We show that, for a cohort born in Britain in a single week in 1958 those reporting aches and pains at age 44 report poorer health, wellbeing and labor market outcomes over the following two decades through age 62. The partial associations between chronic pain and subsequent health and labor market outcomes persist even in the presence of lagged pain measured earlier in life and having controlled for parental and familial background in childhood, as well as a wide range of physical and mental health ailments reported in mid-life. We also find that chronic pain at age 44 in 2002 predicts having Covid nearly two decades later in 2021.

## 3. Pain incidence in the UK and its correlates

Piper et al [21] report the incidence of pain in thirty-eight OECD countries from the Gallup World Polls of 2005–2020 (GWP) based on responses to the following question on pain the previous day:

*Q1. Did you experience the following feelings during a lot of the day yesterday*? *How about physical pain*? *(WP68).*

The UK was included in that sample and in Table 1 we report their table on mean pain incidence. The table confirms earlier findings using data from the International Social Survey Program (ISSP) 2011 showing that pain incidence was higher in the United States than the UK [36]. In the GWP the UK has a mean rate of pain incidence of 21.1%, which ranks 33rd out of 38 OECD countries. Pain is lowest in Ireland (19.2%) and highest in Chile (36.8%). Piper et al [21] found that in a pain regression across all OECD countries, controlling for gender, age, education, marital status, labor market status and year only Poland, Ireland and Japan had less pain than the UK.

*Q1. Experienced physical pain yesterday–yes/no*? *(wp68)*

In Table 2 we run linear regressions for the (0,1) outcome of experiencing physical pain "during a lot of the day yesterday" using GWP pooled over the period 2009–2020. The first

**Table 1. Mean pain rates, Gallup World Poll, 2005–2020 (weighted) %.**

| | | | | | |
|---|---|---|---|---|---|
| Australia | 23.9 | Hungary | 29.7 | Poland | 19.3 |
| Austria | 21.3 | Iceland | 32.1 | Portugal | 33.6 |
| Belgium | 29.6 | Ireland | 19.2 | Slovakia | 27.4 |
| Canada | 27.9 | Israel | 29.8 | Slovenia | 24.9 |
| Chile | 36.8 | Italy | 25.2 | South Korea | 24.4 |
| Colombia | 30.5 | Japan | 19.7 | Spain | 29.3 |
| Czech Republic | 24.1 | Latvia | 22.9 | Sweden | 21.5 |
| Denmark | 24.7 | Lithuania | 24.2 | Switzerland | 23.6 |
| Estonia | 20.6 | Luxembourg | 25.8 | Turkey | 22.6 |
| Finland | 23.7 | Mexico | 28.4 | United Kingdom | 21.1 |
| France | 29.3 | Netherlands | 20.9 | United States | 28.3 |
| Germany | 22.5 | New Zealand | 23 | Total | 25.1 |
| Greece | 28.3 | Norway | 22.5 | | |

Source: [21]

two columns are estimates for the UK. In the first column we control for gender, age and year and we restrict the sample to cases where there were observations available on education and labor force status to be comparable to column 2 (n = 28,284). The model takes the following form:

$$P_i = \beta_0 + \beta'_1 age + \beta_2 male + \beta'_3 estatus + \beta'_4 educ + \beta_5 year \qquad \text{(Eq 1)}$$

where the dependent variable $P_i$ is pain yesterday for individual $i$ as a function of age, specified as a set of categorical dummies, a male dummy employment status dummies, education dummies, and a linear year term.

Column 3 pools data for 166 countries with the UK as the reference category.

Pain incidence is significantly higher among women and rises with age and there is a positive time trend. Higher levels of education are associated with less pain and once again there is a significant and positive time trend. The age and gender effects are similar to column 1. The final column estimates the same equation as in column 2 but now includes the complete set of 166 countries (n = 1.72 million). The coefficients are very similar to those in column 2. We report the coefficients from twelve advanced countries, versus the USA which is excluded. The UK has significantly lower levels of pain, ceteris paribus, than does Canada (+.07), France (+.07), Germany (+.13) and the USA (+.07) but has insignificantly different from the levels of pain compared to Australia, Austria, Finland, Italy, Netherlands, Norway and Sweden and significantly higher than Japan (-.13).

To examine pain incidence by age a little further we run a model using a complete set of year of age dummy variables. In Fig 1A for the years 2005–2020 (n = 35,611) only gender and a time trend are included, and pain rises with age. In Fig 1B we add labor force status and education, which means the sample is restricted to the years 2009–2020, which are the years when the labor market variables are available on a consistent basis (n = 28,284). In both figures we include the complete age range from 15–100 but only plot the data for ages 15–70. This suggests an inverted u-shape in pain as found in Blanchflower [37] for the United States using data from question Q1 above, from the Gallup US Daily Tracker, 2008–2017.

**Table 2. Pain regressions in Gallup World Poll, 2009–2020 (OLS).**

| 25–29 | | .0068 (0.50) | .0393 (2.72) | .0505 (39.19) |
|---|---|---|---|---|
| 30–34 | | .0238 (1.92) | .0669 (5.09) | .0715 (53.94) |
| 35–39 | | .0276 (2.28) | .0682 (5.32) | .0880 (64.45) |
| 40–44 | | .0407 (3.59) | .0853 (7.06) | .1078 (76.12) |
| 45–49 | | .0312 (2.75) | .0772 (6.39) | .1296 (88.38) |
| 50–54 | | .0757 (6.74) | .1123 (9.40) | .1623 108.21) |
| 55–59 | | .0905 (7.97) | .1200 (10.02) | .1858 (116.93) |
| 60–64 | | .1048 (9.22) | .0983 (8.18) | .2086 (128.13) |
| 65–69 | | .0999 (8.59) | .0575 (4.60) | .2282 (126.86) |
| 70+ | | .1474 (14.28) | .0971 (8.54) | .2742 (181.77) |
| Male | | -.0208 (4.31) | -.0057 (1.16) | -.0436 (61.78) |
| Year | | .0044 (4.87) | .0054 (5.89) | .0072 (65.20) |
| Secondary | | | -.0330 (3.85) | -.0713 (81.84) |
| College | | | -.0717 (8.02) | -.1257 (105.84) |
| FT Self-employed | | .0238 (2.37) | .0134 (11.21) | |
| PT | | | .0280 (3.18) | .0181 (12.46) |
| Unemployed | | | .0641 (4.79) | .0242 (15.70) |
| PT wants FT | | | .0555 (4.52) | .0351 (24.19) |
| OLF | | | .1528 (22.05) | .0206 (20.96) |
| Australia | -.0034 (0.67) | | | |
| Austria | -.0080 (1.66) | | | |
| Canada | +.0683 (14.12) | | | |
| Finland | -.0079 (1.57) | | | |
| France | +.0737 (15.58) | | | |
| Germany | +.0135 (3.72) | | | |
| Italy | -.0376 (8.01) | | | |
| Japan | -.1380 (24.40) | | | |
| Netherlands | -.0058 (1.17) | | | |
| Norway | -.0085 (1.52) | | | |
| Sweden | .0028 (0.58) | | | |
| USA | +.0707 (14.62) | | | |
| Constant | | -8.7447 | -10.6684 | -14.3041 |
| N | | 28,284 | 28,284 | 1,722,234 |
| Adjusted R$^2$ | | 0.0198 | 0.042 | 0.0782 |

Equations also include controls for education don't know and not answered. Excluded categories employee and age 15–24. Column 3 excluded country is the UK and the equation includes a total of 165 country dummies. Sample in column 1 same as columns 2 & 3 based on responses being available for labor market status and education.

## 4. The National Child Development Survey (NCDS) and its pain data

The NCDS follows all those born in one week in March in England, Scotland and Wales in 1958 (see http://www.cls.ucl.ac.uk/ncds). Participants are followed throughout their lives, with survey waves taking place at birth in the Perinatal Mortality Study (PMS) and then at ages 7, 11, 16, 23, 33, 42, 44, 46, 50, 55 and 62 making a total of twelve sweeps. The survey includes data from the respondent, their teachers and parents. We make use of data in eleven of the sweeps and exclude NCDS4 taken at age 23 in 1981. In total there are 18,558 respondents including 619 stillbirths and neonatal deaths.

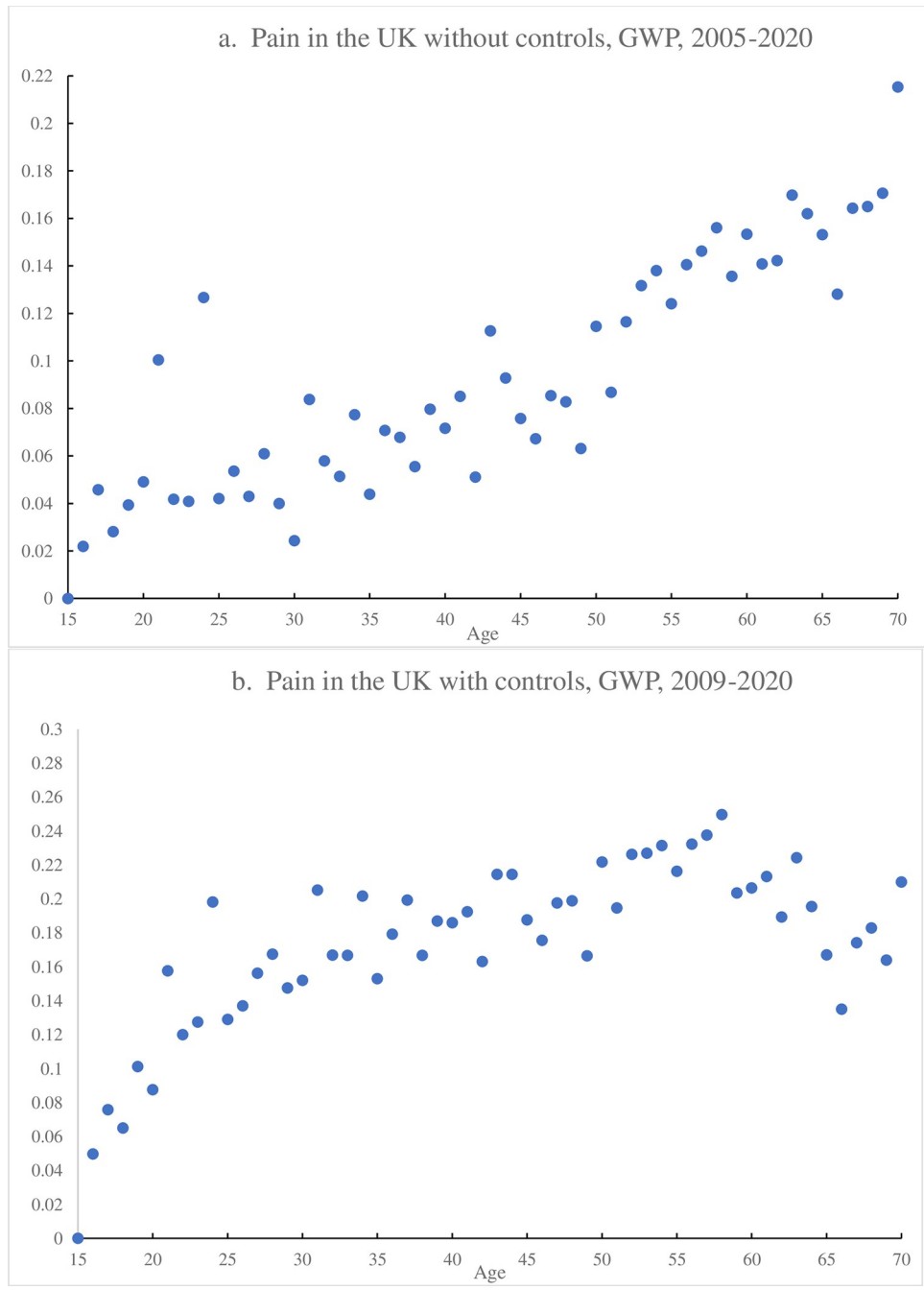

**Fig 1.** a. Pain in the UK without controls, GWP, 2005–2020. b. Pain in the UK with controls, GWP, 2009–2020.

The main pain data we use are taken from the Bio-Medical Survey (BMS) which was conducted when the NCDS respondents were mostly age 44 with the majority of interviews undertaken in 2003. Fieldwork began in September 2002 and was completed at the end of March 2004. The 9377 interviews were undertaken in 2002 = 19.0%; 2003 = 76.1% and 2004 = 4.8%. This meant that at the time of interview 95.2% were age 44; 4.5% were age 45 and 0.3% were age 46. The target sample comprised of 12,037 cohort members living in Scotland, England or Wales who responded to NCDS4, 5 or 6 (when they were aged 23, 33 and 42), and involved

nurse-interviewers taking a number of biomedical measurements, including: near, distance and stereo vision; hearing; lung function; blood pressure and pulse, height and weight; and waist and hip measurements. A short mental health interview was also administered, and samples of blood and saliva were taken. Levels of co-operation with the survey were high, with some 9,400 cohort members taking part, and only a small minority declining to provide samples of blood and saliva.

The appendix reports the questions used to determine pain in the BMS. Respondents were first asked whether in the prior month they had experienced an ache or pain lasting for one day or longer. If they answered in the affirmative, they were then asked if they had been aware of it for more than three months. We identify those saying they have experienced pain for at least three months who we call those in *chronic pain* and those who say pain lasted for less than that as being in *short pain*. They have means of 40.6% and 13.0% respectively. Overall, then, at age 44 around half the sample reported experiencing pain lasting one day or more.

The pain data in the BMS were validated by Macfarlane et al [38] against those in other data files including against a UK Biobank sample of half a million ages 40–69 versus the 9377 respondents in NCDS. They found prevalence of pain shown in Table 3.

Macfarlane and co-authors found that chronic pain decreased with income and those will less education were more likely to report chronic pain. In relation to employment, chronic pain was least common among those in paid employment (39.8%) and those doing unpaid or voluntary work (42.3%), while four-fifths (78.9%) of persons who were unable to work because of ill-health reported chronic pain.

We link the pain data from the BMS to the NCDS data which reports characteristics of the respondent at seven other sweeps 2–4 (ages 7, 11 and 16) and 5–9 (ages 33, 42, 46, 50 and 55) plus the 2021 NCDS Covid study panel 3 (age 62) and the Perinatal Mortality Study at birth in 1958, which also includes data on their parents. The sweeps were as follows with age in parentheses: PM (0) 1958; NCDS1 (7) 1965; NCDS2 (11) 1969; NCDS3 (16), 1974; NCDS4 (23) 1991; NCDS5 (33) 1991; NCDS6 (42) 2000; BMS (44) 2002; NCDS7 (46) 2004; NCDS8 (50) 2008; NCDS9 (55) 2013; NCDS Covid (62) 2020. Sample sizes are PMS = 7,415; NCDS1 = 15,425; NCDS2 = 15,337; NCDS3 = 14,654; NCDS4 = 12,537; NCDS5 = 11,469; NCDS6 = 11,419; NCDSBMS = 9,377; NCDS7 = 9,534; NCDS8 = 9,790; NCDS9 = 9,137; NCDS Covid = 6,809. The overall sample size is 18,558: 16,370 were alive after 28 days. Emigrants and those born in the week that were not included in the PMS were subsequently added and of course, some respondents emigrated. At the time of the BMS 1116 had emigrated and there had been 1193 deaths including the deaths within 28 days of birth. By age 55 total deaths had risen to 1,659 deaths. Some respondents dropped out of the sample and then came back. For example, 8,558 provided responses to both the BMS and NCDS7 surveys; 819 did not respond to the BMS but did respond to NCDS7, while 976 who did no responded to the BMS did not respond to NCDS7, so the sample size rose by 157 from 9377 to 9534. Of interest is the distribution of responses to the ten surveys, including BMS through NCDS9. Overall, a quarter

**Table 3. Pain prevalence (%).**

|  | UK Biobank | NCDS |
|---|---|---|
| Any pain | 62.8 | 53.3 |
| Chronic pain | 41.6 | 40.9 |
| Shoulder/neck pain | 23.5 | 20.4 |
| Back pain | 27.4 | 26.2 |
| Hip pain | 7.4 | 15.4 |
| Knee pain | 17.1 | 19.1 |

responded to all ten (25.5%); 15.6% to nine sweeps and 9.7% to 8 sweeps. Two thirds of the sample responded to at least 5 sweeps.

As shown in Table 4 we have available to us nine pain variables. The first three are reported by someone other than the respondent. Two reports are available on abdominal pain from the respondent's mother at ages seven and eleven. Then the school reported at age sixteen on whether they complained of aches and pains. The respondents reported on back pain at age thirty-three, forty-four (or myalgic encephalomyelitis or chronic fatigue syndrome), and fifty-five and on bodily pain over the prior month at age 50. Chronic fatigue syndrome or ME is a debilitating illness that affects about 250,000 children and adults in the UK alone who are sometimes left bed-bound for years on end [39]. At age 42 in ncds6 (variable = *backme1*) 2542 cases report persistent back ache or sciatica, 72 report ME and 8761 say none of these.

The main two variables we focus on in this paper are more general and are taken from the BMS which was in the field from 2002–2004, where respondents were asked '*during the past month have you had any aches or pains which have lasted for one day or longer*'. Respondents were then asked: '*have you been aware of it for more than three months*?' We thus construct two variables one identifying pain of 1 day to under three months–**short pain**—and our most important variable–**chronic pain**–which is pain lasting for more than 3 months.

In what follows we primarily examine the impact of chronic and short pain, taken at age 44 in 2002 on subsequent outcomes. These outcomes include

a.  Bodily pain at age 50 (Table 8).

b.  Backache at age 55 (Table 10).

c.  Whether the respondent was working at age 55 (Table 11).

d.  Depression, emotional and psychiatric problems at age 55 (Table 12).

e.  General health at age 55 (Table 13).

f.  Attitudes to life at age 55 (Table 14)

g.  Life satisfaction and Malaise scores at age 50 (Table 15)

h.  Hours of sleep and difficulty falling asleep (Table 16)

i.  Life satisfaction and malaise scores at age 62 (Table 17)

j.  Catching covid at age 62 (Table 17)

**Table 4. Pain Variables in NCDS.**

| Pain variables | N | | | |
|---|---|---|---|---|
| Pain age 7 | NCDS1 | 14,519 | Periodic abdominal pain reported by parent. | |
| Pain age 11 | NCDS2 | 13,626 | Periodic abdominal pain reported by parent. | |
| Pain age 16 | NCDS3 | 12,423 | Complains of aches or pains reported by teachers/school. | |
| Pain age 33 | NCDS5 | 11,325 | Ever had back pain lasting >1 day. | |
| Pain age 44 | NCDS6 | 11,375 | Ever had persistent back pain or ME. | |
| Pain age 50 | NCDS8 | 8,750 | How much bodily pain past 4 weeks. | |
| Pain at age 55 | NCDS9 | 9,021 | Health problems since last interview -backache, sciatica, or disc prolapse. | |
| Chronic pain age 44 | BMS | 8,433 | Ache or pain during last month lasting >3 months. | |
| Short pain at age 44 | BMS | 8,433 | Ache or pain during last month lasting >1 day <3 months. | |

**Table 5. Chronic Pain in BMS (age 44) and characteristics at age 46 and at birth %.**

| | | | |
|---|---|---|---|
| All | 40.9 | Smokes occasionally | 44.4 |
| Male | 40.4 | No qualifications | 49.5 |
| Female | 41.4 | Bad O-levels | 44.6 |
| North | 41.9 | Good O-level, 1 A level | 39.5 |
| Yorkshire & Humber | 41.5 | 2 or more | 36.3 |
| East Midlands | 42.9 | Sub-degree | 41.8 |
| East Anglia | 40.7 | Degree | 35.6 |
| South East | 40.7 | Higher degree | 27.1 |
| South West | 38.7 | Excellent health | 27.4 |
| West Midlands | 42 | Good health | 39.5 |
| North West | 41.4 | Fair health | 70.1 |
| Wales | 41.4 | Poor health | 79.3 |
| Scotland | 37.5 | owns home outright | 36.3 |
| Worker | 38.4 | Owning with mortgage | 39.8 |
| Unemployed | 51.4 | Renter | 51.8 |
| OLF | 58.8 | Rent free | 39.3 |
| Never smoked | 37.2 | Low life satisfaction score <8 | 46.3 |
| Used to smoke not now | 42.5 | High life satisfaction score 8–10 | 37.9 |
| Socio-economic group mother's husband in 1958 | | | |
| Skilled workers | 41.7 | | |
| Semi-skilled | 45.4 | | |
| Unskilled worker | 44.5 | | |
| Armed forces | 44 | | |
| Admin, prof, managers | 35.4 | | |
| Shopkeepers | 39.4 | | |
| Clerical workers | 38.1 | | |
| Shop assistants | 42.1 | | |
| Personal service | 43.3 | | |
| Foremen | 35.9 | | |
| Farmers | 38.6 | | |
| Farm workers | 43.6 | | |
| Higher administration | 33.2 | | |
| Single, no husband | 41 | | |

*Note*: Chronic pain data from BMS, and the characteristics data other than gender from NCDS7 and the PMS, 1958

We show chronic pain predicts a variety of objective and subjective outcomes years later. We show that our chronic pain variable, observed at age 44 significantly,

1) raises the probability of having bodily pain at 50 and backache at 55 years,

2) lowers life satisfaction at age 46 and 50 and general health and attitudes to life at age 55,

3) raises depression at age 55 and unhappiness measured by the Malaise score at age 50,

4) lowers the probability of working a decade later at age 55,

5) reduces hours of sleep and makes it more likely that sleep will be troubled,

5) and finally, raises the probability of having had Covid eighteen years later.

**Table 6. Attrition and Chronic Pain in the BMS (OLS), 1958–2004.**

|  | Attritted | Chronic Pain at 44 |  |  |  | Weighted |
|---|---|---|---|---|---|---|
| Pain at age 16 |  |  |  |  |  |  |
| Applies somewhat | .0962 (5.36) | .0918 (3.53) | .0789 (2.79) | .0716 (2.41) | .0512 (1.66) | .0931 (3.50) |
| Certainly applies | .1447 (4.40) | .1278 (2.54) | .1617 (3.00) | .1648 (2.80) | .1369 (2.20) | .1270 (2.48) |
| Missing | .0662 (5.64) | .0324 (1.98) | .0335 (1.90) | .0308 (1.67) | .0314 (1.64) | .0324 (1.96) |
| Pain at age 11 |  |  | .0534 (2.53) | .0530 (2.55) | .0471 (2.20) |  |
| Pain at age 7 |  |  |  | .0305 (1.71) | .0267 (1.45) |  |
| Back pain at age 33 |  |  |  |  | .1119 (8.18) |  |
| Back pain at age 42 |  |  |  |  | .2358 (14.58) |  |
| Semi-skilled | .0300 (1.92) | .0357 (1.67) | .0428 (1.87) | .0327 (1.37) | .0070 (0.29) | .0343 (1.58) |
| Unskilled worker | .0899 (5.79) | .0313 (1.40) | .0307 (1.27) | .0292 (1.15) | .0075 (0.28) | .0280 (1.24) |
| Armed forces | .0336 (1.37) | .0343 (1.03) | .0232 (0.62) | .0050 (0.12) | .0051 (0.12) | .0311 (0.93) |
| Admin, prof, managers | -.0624 (3.87) | -.0638 (3.12) | -.0658 (3.04) | -.0763 (3.39) | -.0753 (3.27) | -.0659 (3.30) |
| Shopkeepers | -.0644 (2.02) | -.0070 (0.18) | -.0052 (0.12) | -.0275 (0.62) | -.0414 (0.91) | -.0094 (0.23) |
| Clerical workers | -.0345 (1.54) | -.0192 (0.67) | -.0187 (0.62) | -.0211 (0.67) | .0149 (0.47) | -.0184 (0.64) |
| Shop assistants | -.0539 (2.39) | .0101 (0.35) | .0154 (0.50) | .0018 (0.06) | .0146 (0.44) | .0103 (0.35) |
| Personal service | .0289 (0.73) | .0522 (0.96) | .0520 (0.89) | .0730 (1.17) | .0638 (0.99) | .0550 (0.99) |
| Foremen | -.0108 (0.36) | -.0566 (1.41) | -.0543 (1.24) | -.0574 (1.29) | -.0506 (1.13) | -.0583 (1.49) |
| Farmers | -.0294 (0.87) | -.0288 (0.66) | -.0202 (0.44) | -.0339 (0.73) | -.0744 (1.58) | -.0262 (0.60) |
| Farm workers | -.0722 (2.81) | .0155 (0.48) | .0050 (0.15) | -.0055 (0.16) | -.0381 (1.06) | .0155 (0.48) |
| Higher admin etc. | -.1004 (4.43) | -.0840 (3.00) | -.0815 (2.74) | -.0790 (2.53) | -.07795 (2.46) | -.0831 (3.08) |
| Single, no husband | .1190 (4.26) | -.0337 (0.81) | -.0475 (1.03) | -.0716 (1.42) | -.0612 (1.16) | -.0345 (0.83) |
| Male | .0537 (5.19) | -.0125 (1.11) | -.0124 (1.01) | -.0117 (0.92) | -.0092 (0.70) | -.0124 (1.10) |
| Region—1958 | Yes | Yes | Yes | Yes | Yes | Yes |
| Constant | 0.3827 | 0.4193 | 0.4184 | 0.4075 | 0.3135 | 0.4185 |
| N | 13,859 | 7,685 | 6,637 | 6,040 | 5,311 | 7,685 |
| Adjusted $R^2$ | 0.0245 | 0.0048 | 0.0063 | 0.006 | 0.072 | 0.0073 |

T-statistics in parentheses. Column 6 is weighted using inverse probability weights from column 1. Excluded skilled workers.

Mother's husband's 's social class also includes two variables for missing not reported

In some instances, we find that as we move further away in time from the observation of chronic pain at age 44 its significance wanes, and more recent experience of pain becomes significant. This occurs in Table 17 for the Malaise score at age 62 –which indicates tendency towards depression. In this case pain at 55 and at 50 are both significant predictors. This is also true for life satisfaction at age 62 even when three lagged life satisfaction variables are included as controls.

**Table 7. Pain in NCDS at Age 50 (cell percentages).**

|  | Had chronic pain at age 44 | No qualifications at age 50 | Degree of Higher at age 50 |
|---|---|---|---|
| Pain at age 50: |  |  |  |
| None | 21% | 24% | 31% |
| Very mild | 35% | 21% | 35% |
| Mild | 47% | 18% | 17% |
| Moderate | 60% | 22% | 12% |
| Severe | 75% | 11% | 4% |
| Very severe | 84% | 3% | 1% |

**Table 8. Bodily pain in NCDS8 at age 50 (OLS).**

| Chronic pain at 44 | 1.0076 (32.10) | .8533 (25.17) | .7593 (22.94) | .7568 (22.84) | .7552 (21.78) |
|---|---|---|---|---|---|
| Short pain at 44 | .4239 (9.04) | .3567 (7.35) | .3566 (7.61) | .3556 (7.57) | .3577 (7.28) |
| Back pain at age 42 | | .4199 (10.86) | .3814 (10.21) | .3789 (10.12) | .3671 (9.39) |
| Back pain at age 33 | | .1942 (6.040 | .1843 (5.94) | .1809 (5.82) | .1784 (5.49) |
| Pain at age 16 | | | | | |
| Applies somewhat | | | | | .1867 (2.47) |
| Certainly applies | | | | | .3719 (2.51) |
| Missing | | | | | -.0233 (0.52) |
| Male | -.2109 (7.18) | -.2297 (7.57) | -.1998 (5.94) | -.2002 (5.94) | -.2071 (5.43) |
| Region at 50 | Yes | Yes | Yes | Yes | Yes |
| Personal controls | No | No | Yes | Yes | Yes |
| Father's social class at 0 | No | No | No | Yes | Yes |
| Constant | 2.1771 | 2.0831 | 2.2186 | 2.2099 | 2.2013 |
| Adjusted $R^2$ | 0.1367 | 0.1653 | 0.2212 | 0.2211 | 0.2232 |
| N | 6,860 | 6,079 | 6,079 | 6,079 | 5,562 |

Notes: personal controls are labor force status and education in NCDS8. T-statistics in parentheses.

(Others [40, 41] have used the NCDS malaise variable. It is a 9-item measure with yes/no responses and is used as a measure of psychological distress).

Initially though we focus on sample attrition in relation to the responses on short and chronic pain in the BMS surveys. We also look at the determinants of chronic pain including the social class of the mother's husband at the time of the respondent's birth in March 1958. We only have information on the mother's husband, reported by the NCDS respondent's mother and we do not know directly if this person is the father.

Our strategy here is to estimate a series of equations which include a chronic pain variable as a control and other lagged pain variables, that exploit the longitudinal nature of the data file. The changing wording of the questions means that simply including person fixed effects is not feasible. We show that variations in specification have little impact on our results. Chronic pain at age 44 impacts many variables years later; it suggests it represents other comorbidities impacting wellbeing and reflecting vulnerabilities, which make those with chronic pain more susceptible to catching covid nearly two decades later at age 62.

**Table 9. Incidence (%) of back pain symptoms in NCDS at Age 55.**

| | | | |
|---|---|---|---|
| All | 26 | Degree | 21.4 |
| Males | 24.9 | Higher degree | 18.8 |
| Females | 27 | Poor health | 58.5 |
| Workers | 23.4 | Fair health | 41.4 |
| Unemployed | 26.3 | Good health | 27.4 |
| Permanent sick | 57.2 | Very good health | 18.4 |
| Looking after family | 33 | Excellent health | 10.9 |
| Retired | 23.6 | Living comfortably | 21.6 |
| No qualifications | 32 | Doing all right | 23.5 |
| CSEs 2–5 | 27.7 | Just about getting by | 29.9 |
| GCSE | 26.2 | Finding it quite difficult to pay bills | 34 |
| A-levels | 22.7 | Finding it very difficult to pay ills | 49.7 |

**Table 10. Backache, sciatica, disk prolapse and pain lasting > one day–NCDS9 at 55 (OLS).**

| | | | | |
|---|---|---|---|---|
| Bodily pain very mild at 50 | .0773 (6.19) | .0562 (4.08) | .0356 (2.53) | .0402 (2.43) |
| Bodily pain mild at 50 | .1767 (12.07) | .1243 (7.47) | .1046 (6.12) | .1036 (5.19) |
| Bodily pain moderate at 50 | .2834 (19.00) | .2281 (13.38) | .1884 (10.65) | .1804 (8.54) |
| Bodily pain severe at 50 | .3725 (16.34) | .2815 (10.81) | .2369 (8.81) | .2559 (8.34) |
| Bodily pain very severe at 50 | .4568 (10.64) | .3520 (7.02) | .2777 (5.24) | .2653 (4.30) |
| Chronic pain at 44 | | .1507 (12.40) | .1074 (8.50) | .1040 (6.97) |
| Short pain at 44 | | .0605 (3.58) | .0561 (3.26) | .0539 (2.70) |
| Back pain at age 42 | | | .1934 (14.03) | .1890 (11.6) |
| Back pain at age 33 | | | .0844 (7.38) | .0923 (6.86) |
| Pain at age 16 | | | | |
| Applies somewhat | | | | .0029 (0.009) |
| Certainly applies | | | | .0501 (0.76) |
| Missing | | | | .0232 (1.25) |
| Pain at 11 | | | | -.0114 (0.51) |
| Pain at 7 | | | | .0212 (1.08) |
| Male | .0093 (0.88) | .0130 (1.11) | .0038 (0.32) | .0369 (0.97) |
| Personal controls | Yes | Yes | Yes | Yes |
| Region at 55 | Yes | Yes | Yes | Yes |
| Constant | 0.0817 | 0.0516 | 0.1461 | 0.0372 |
| Adjusted $R^2$ | 0.1108 | 0.1264 | 0.1763 | 0.1767 |
| N | 7,460 | 6,098 | 5,461 | 3,986 |

Notes: education, labor force status and difficulty paying bills. Excluded pain at 50 = none.

## 5. Results

In Table 5 we report how chronic pain observed in the BMS at age 44 relates to the characteristics of these individuals in the NCDS7 interviews at age 46. 80.1% of 9534 interviews for NCDS7 were conducted in 2004 and 1,896 in 2005 and 96% of respondents were age 46 at

**Table 11. Paid Work in NCDS9 at 55.**

| | | | | |
|---|---|---|---|---|
| Bodily pain very mild at 50 | -.0089 (0.81) | -.0042 (0.36) | -.0011 (0.11) | .0014 (0.14) |
| Bodily pain mild at 50 | -.0048 (0.38) | -.0002 (0.01) | .0035 (0.30) | .0013 (0.11) |
| Bodily pain moderate at 50 | -.0942 (7.25) | -.0886 (6.01) | -.0451 (3.69) | -.0417 (3.37) |
| Bodily pain severe at 50 | -.2541 (13.18) | -.2322 (10.57) | -.0722 (3.87) | -.0656 (3.46) |
| Bodily pain very severe at 50 | -.4834 (13.29) | -.5129 (12.33) | -.1575 (4.40) | -.1729 (4.78) |
| Chronic pain at 44 | | -.0290 (2.75) | -.0054 (0.62) | .0018 (0.21) |
| Short pain at 44 | | .0093 (0.64) | .0044 (0.37) | .0060 (0.49) |
| Male | .0824 (9.75) | .0760 (8.22) | .0124 (1.44) | .0077 (0.86) |
| Labor force status age 50 | No | No | Yes | Yes |
| Labor force status age 46 | No | No | No | Yes |
| Personal controls | Yes | Yes | Yes | Yes |
| Constant | 0.7028 | 0.6229 | 0.617 | 0.8774 |
| Adjusted $R^2$ | 0.1183 | 0.1131 | 0.3956 | 0.4023 |
| N | 7473 | 6108 | 6108 | 5850 |

Notes: all equations include controls for region at age 55. Personal controls are financial situation, labor force status and education in NCDS9 at age 55. Mean = 81.2%

**Table 12. Depression, emotional and psychiatric problems since last interview NCDS9 at 55.**

| | | | | |
|---|---|---|---|---|
| Bodily pain very mild at 50 | .0248 (2.49) | .0323 (2.13) | .0226 (1.96) | .0145 (1.28) |
| Bodily pain mild at 50 | .0548 (4.69) | .0454 (3.45) | .0371 (2.65) | .0262 (1.91) |
| Bodily pain moderate at 50 | .0918 (7.70) | .0774 (5.72) | .0755 (5.21) | .0522 (3.66) |
| Bodily pain severe at 50 | .0802 (4.41) | .0731 (3.54) | .0674 (3.06) | .0336 (1.55) |
| Bodily pain very severe at 50 | .1414 (4.13) | .1350 (3.40) | .1440 (3.32) | .1215 (2.82) |
| Chronic pain at 44 | | .0348 (3.61) | .0295 (2.86) | .0122 (1.19) |
| Short pain at 44 | | .0171 (1.28) | .0104 (0.74) | .0019 (0.14) |
| Back pain at age 42 | | .0252 (2.23) | .0183 (1.66) | |
| Back pain at age 33 | | .0235 (2.51) | .0190 (2.07) | |
| Male | -.0419 (4.92) | -.0300 (3.72) | -.0320 (3.88) | -.0307 (3.14) |
| Fatigue score at 44 | | | | .0164 (4.07) |
| Concentration score at 44 | | | | .0160 (1.92) |
| Sleep score at 44 | | | | .0090 (2.02) |
| Irritability score at 44 | | | | .0141 (2.59) |
| Depression score at 44 | | | | .0121 (1.37) |
| Depressive ideas score at 44 | | | | .0228 (2.58) |
| Anxiety score at 44 | | | | .0211 (2.74) |
| Phobias score at 44 | | | | .0450 (5.04) |
| Panic score at 44 | | | | .0177 (1.15) |
| Personal controls | Yes | Yes | Yes | Yes |
| Region at 55 | Yes | Yes | Yes | Yes |
| Constant | 0.0678 | 0.053 | 0.0467 | 0.0168 |
| Adjusted $R^2$ | 0.0242 | 0.1362 | 0.1304 | 0.1688 |
| N | 8,266 | 6,094 | 5,458 | 5,448 |
| Notes: personal controls are region, labor force status, education and financial status in NCDS9 at age 55. All mental health scores at age 42 | | | | |
| Means | | | | |
| Fatigue score at 44 | 0.831 | | | |
| Concentration score at 44 | 0.247 | | | |
| Sleep score at 44 | 0.731 | | | |
| Irritability score at 44 | 0.573 | | | |
| Depressive ideas score at 44 | 0.235 | | | |
| Anxiety score at 44 | 0.275 | | | |
| Phobias score at 44 | 0.18 | | | |
| Panic score at 44 | 0.044 | | | |

time of interview. The characteristics are taken from NCDS7 because most are not reported at the BMS, which primarily collected biomedical data. Of the 8,433 observations in the BMS 40.9% reported having chronic pain. An additional 690 respondents to the BMS did not fill out the self-completion pain questionnaire while 102 had missing values.

The table shows chronic pain was higher in females than males which is one possible reason why Blanchflower and Bryson [42, 43] find females had higher negative affect than men. Chronic pain is highest in the raw data among the least educated, smokers and the unemployed and especially so among those who were out of the labor force (OLF). It is highest in the East Midlands and lowest in Scotland. Those with chronic pain also had relatively low life satisfaction scores.

We are also able to identify social class of the mother's husband at the time of the respondent's birth in 1958, noting of course that social class is a variable that changes over time. We

**Table 13. General health in NCDS9 at 55 (OLS).**

| | | | | |
|---|---|---|---|---|
| Health at 50 fair | .5974 (11.09) | .3040 (5.61) | .2758 (4.47) | .2450 (3.84) |
| Health at 50 good | 1.2861 (24.82) | .8551 (15.89) | .8062 (13.13) | .6889 (10.67) |
| Health at 50 very good | 1.7694 (33.67) | 1.3096 (23.94) | 1.2515 (20.04) | 1.0447 (15.74) |
| Health at 50 excellent | 2.3406 (4.65) | 1.8325 (32.13) | 1.7867 (27.51) | 1.4653 (21.07) |
| Bodily pain very mild at 50 | -.0996 (4.12) | -.1018 (4.36) | -.0727 (2.85) | -.0803 (3.15) |
| Bodily pain mild at 50 | -.1699 (5.89) | -.1584 (5.69) | -.0967 (3.09) | -.0935 (2.99) |
| Bodily pain moderate at 50 | -.2897 (9.58) | -.2485 (8.50) | -.1845 (5.58) | -.1632 (4.92) |
| Bodily pain severe at 50 | -.4578 (10.03) | -.2982 (6.70) | -.2203 (4.38) | -.1914 (3.80) |
| Bodily pain very severe at 50 | -.7342 (8.70) | -.4362 (5.28) | -.3271 (3.39) | -.3007 (3.11) |
| Chronic pain at 44 | | | -.0884 (3.92) | -.0489 (2.16) |
| Short pain at 44 | | | .0039 (0.13) | .0251 (0.81) |
| Health at 46 fair | | | | -.0955 (0.99) |
| Health at 46 good | | | | .0163 (0.17) |
| Health at 46 very good | | | | .2386 (2.56) |
| Health at 46 excellent | | | | .5305 (5.58) |
| Male | .0046 (0.25) | -.0151 (0.76) | -.0304 (1.40) | -.0255 (1.18) |
| Mental Health score at 44 | | | -.0123 (4.96) | -.0075 (2.97) |
| Personal controls | No | Yes | Yes | Yes |
| Region at age 55 | Yes | Yes | Yes | Yes |
| Constant | 1.8768 | 2.4762 | 2.5567 | 2.4234 |
| Adjusted $R^2$ | 0.4357 | 0.484 | 0.4824 | 0.3186 |
| N | 7,573 | 7,455 | 6,090 | 5,831 |

Notes: all equations include region at age 55. Personal controls are financial situation, labor force status and education in NCDS9 at age 55. Rating of general health at age 55: Poor = 6.0%; fair = 14.0%; good = 32.3%; very good = 34.2% and excellent = 13.5%. Rating of general health at age 50: Poor = 5.7%; fair = 12.7%; good = 29.2%; very good = 33.0% and excellent = 19.4%. %. Rating of general health at age 44: Poor = 1.9%; fair = 5.5%; good = 16.0%; very good = 45.2% and excellent = 31.5%.

find that chronic pain is higher the less skilled is the occupation, and lower for professional and administrative occupations and farmers.

## 5.1. Predictors of sample attrition and chronic pain at age 44

In Table 6 we use linear estimation to model the probability of attrition in the Biomedical Survey (BMS) at age 44 using linear estimation. There are a total of 18,558 individuals in the NCDS including deaths at birth. The attrition variable is set to zero for the 8433 cases where the respondent reported their chronic pain status at age 44 and was set to one in the 792 cases that responded to the BMS but did not have valid responses. We also set to missing cases that were stillbirths (n = 367), neonatal deaths (n = 252) or had died by NCDS7 (n = 667) or had emigrated by the time of the BMS (n = 1236). This leaves 7,603 cases set to one that attritted (47.4%) and 8,433 cases that did not.

The attrition model takes the following form:

$$Attrit_{44} = \beta_0 + \beta_1 pain_7 + \beta_2 pain_{11} + \beta_3 pain_{33} + \beta_4 pain_{42} + \beta_5 male + \beta_5' Xrb_0 + \beta_5' Xfb_0 \text{ (Eq 2)}$$

where $Attrit_{44}$ is attrition at age 44, as a function of the individual's pain lagged at ages 42, 33, 11 and 7, being male, region at birth and a vector capturing mother's husband's social class at birth. The i subscripts for individual i are omitted from this and subsequent NCDS models, but these are within-person estimators tracking the same people in the birth cohort over time.

**Table 14. Attitudes in NCDS9 at age 55.**

|  | Full of energy these days | Full of energy these days | Life is full of opportunities | Life is full of opportunities | Future looks good for me | Future looks good for me |
|---|---|---|---|---|---|---|
| Bodily pain very mild at 50 | -.1129 (4.44) | -.1046 (3.99) | -.0597 (2.41) | -.0581 (2.28) | -.0506 (2.26) | -.0469 (2.02) |
| Bodily pain mild at 50 | -.2043 (6.77) | -.1761 (5.56) | -.1372 (4.68) | -.1174 (3.81) | -.0678 (2.56) | -.0462 (1.65) |
| Bodily pain moderate at 50 | -.3727 (12.10) | -.3457 (10.57) | -.1841 (6.15) | -.1656 (5.21) | -.1336(4.92) | -.1145 (3.96 |
| Bodily pain severe at 50 | -.4208 (8.87) | -.3881 (7.78) | -.1255 (2.72) | -.0978 (2.02) | -.1414 (3.38) | -.1178 (2.67) |
| Bodily pain very severe at 50 | -.6404 (6.98) | -.6385 (6.62) | -.1615 (1.80) | -.1372 (1.46) | -.0658 (0.81 | -.0415 (0.49) |
| Chronic pain at 44 |  | -.0864 (3.70) |  | -.0449 (1.98) |  | -.0360 (1.74) |
| Short pain at 44 |  | -.0700 (2.17) |  | -.0068 (0.22) |  | -.0017 (0.06) |
| Mental Health score at 44 | -.0266 (10.88) | -.0258 (10.08) | -.0178 (7.47) | -.0179 (8.68) | -.0194 (8.98) | -.0193 (8.50) |
| Personal controls | Yes | Yes | Yes | Yes | Yes | Yes |
| Constant | 3.3057 | 3.3402 | 3.2914 | 3.3069 | 3.6176 | 3.6289 |
| Adjusted $R^2$ | 0.1716 | 0.1716 | 0.1475 | 0.1533 | 0.2189 | 0.2224 |
| N | 6,487 | 6,083 | 6,482 | 6,077 | 6,463 | 6,061 |

Notes: personal controls are region; financial situation and education dummies in NCDS9 at age 55. Mental health score at age 46.

Mean energy 2.94; life full = 3.08 and future 3.24

Questions

'I feel full of energy these days'—never = 7.7%; not often = 19.4%; sometimes = 45.5% and often = 27.4%

'I feel that life is full of opportunities'—never = 5.3%; not often = 17.0%; sometimes = 44.3% and often = 33.4%

'I feel that the future looks good for me'—never = 4.2%; not often = 15.1%; sometimes = 42.8% and often = 42.1%

Pain at age 16 is taken from teacher responds to the following question (*n2317*).

*"Below is a series of descriptions of behaviour often shown by school children. Please ring the appropriate number in each case to show the degree to which the study child exhibits the*

**Table 15. Life satisfaction and malaise scores in NCDS7 at 46 and NCDS8 at 50 (OLS).**

|  | Life satisfaction at 46 | Life satisfaction at 50 | Life satisfaction at 50 | Malaise score at 50 | Malaise score at 50 | Malaise score at 50 |
|---|---|---|---|---|---|---|
| Chronic pain at 44 | -.1483 (4.61) | -.1589 (49.31) | -.2118 (5.41) | .5943(13.30) | .4921 (11.15) | .4780 (10.76) |
| Short pain at 44 | -.0930 (1.96) | -.0665 (1.16) | -.0287 (0.51) | .2815 (4.27) | .2255 (3.48) | .1956 (3.01) |
| Male | -.1285 (3.75) | .0143 (0.35) | .0256 (0.63) | -.4865 (10.41) | -.5070 (10.93) | -.5285 (11.42) |
| Life satisfaction at 42 | .3329 (40.38) |  | .2308 (20.78) |  |  | -.1279 (9.96) |
| Life satisfaction at 46 |  | .6347 (49.31) | .5071 (36.03) |  | -.3570 (24.59) | -.2799 (17.21) |
| Region | Yes | Yes | Yes | Yes | Yes | Yes |
| Personal controls | Yes | Yes | Yes | Yes | Yes | Yes |
| Constant | 5.33 | 2.6666 | 1.8983 | 1.4539 | 4.2305 | 4.5916 |
| Adjusted $R^2$ | 0.2331 | 0.3031 | 0.3449 | 0.0654 | 0.1896 | 0.1854 |
| N | 7,517 | 6,996 | 6,808 | 7,379 | 6991 | 6,803 |

Notes: personal controls in columns 2 and 3 are labor force status and education dummies in NCDS8 at age 50. Malaise score from 0–9 the components of the malaise score are based on yes (= 1) and no (= 0) answers that are summed across these 9 questions outlied in the text.

The NCDS8 malaise score has a mean of 1.49 with 44% responding no to all questions. It is distributed as follows by number of positive responses—0 = 44.2%; 1 = 20.5%; 2 = 12.5%; 3 = 8.2%; 4 = 5.1%; = 3.7%; 6 = 2.7%; 7 = 1.7%; 8 = 1.2%; 9 = 0.3%.

**Table 16. Hours of sleep and short sleep and problems falling asleep in NCDS8 at 50.**

| | Hours of sleep | Hours of sleep | Short sleep (<7hrs) | Short sleep (<7hrs) | How often woken and had trouble falling asleep in past 4 weeks | |
|---|---|---|---|---|---|---|
| Chronic pain at 44 | -.1539 (4.95) | -.1359 (4.19) | .0628 (5.46) | .0588 (4.79) | .2447 (7.19) | .2113 (5.85) |
| Short pain at 44 | -.0843 (1.88) | -.0815 (1.75) | .0475 (2.84) | .0461 (2.63) | .1224 (2.49) | .1057 (2.06) |
| Back pain at age 33 | | | -.0767 (2.57) | | .0203 (1.81) | .0885 (2.68) |
| Male | -.0857 (2.68) | -.0521 (1.66) | .0004 (0.04) | -.0064 (0.51) | -.2950 (8.41) | -.3127 (8.42) |
| Mental Health score at 44 | -.0438 (12.80) | -.0429 (11.86) | .0161 (12.86) | .0161 (11.99) | .0829 (22.17) | .0821 (20.61) |
| Region | Yes | Yes | Yes | Yes | Yes | Yes |
| Personal controls | Yes | Yes | Yes | Yes | Yes | Yes |
| Constant | 7.0312 | 7.0309 | 0.2586 | 0.2578 | 2.5135 | 2.5034 |
| Adjusted R$^2$ | 0.0205 | 0.0441 | 0.038 | 0.0382 | 0.1458 | 0.0318 |
| N | 6,791 | 6,122 | 7,423 | 6,676 | 6,859 | 6,192 |

Notes: personal controls are labor force status and education dummies in NCDS8 at age 50. Short sleep is versus normal sleep of 7–9 hours so those with > = 10 hrs are dropped in columns3 and 4. Mean sleep hours = 6.91

Question

How often have you woken and had trouble falling asleep in past 4 weeks—none of the time = 18.7%; a little of the time = 30.1%; some of the time = 19.0%; a good bit of the time = 10.0%; most of the time = 7.5%; and all of the time = 4.3%

*behaviour described. Please complete on the basis of the child's behaviour in the past 12 months. . .often complains of aches and pains."*

The variable (n = 14,653 cases) is coded as 1 = does not apply (77%), 2 = applies somewhat (6%), 3 = certainly applies (2%) and 9 = not answered (15%)".

**Table 17. Covid, health and well-being, covid sweep 3 survey at 62, February-March 2021.**

| | Hours of sleep | Life satisfaction | Malaise score | Had Covid | Had Covid |
|---|---|---|---|---|---|
| Back pain at 55 | -.0763 (1.69) | -.1853 (3.06) | .2323 (3.90) | -.0011 (0.09) | |
| Chronic pain at 44 | -.0824 (1.92) | | | .0280 (2.42) | .0275 (2.72) |
| Short pain at 44 | -.0739 (1.28) | | | .0235 (1.50) | .0268 (1.82) |
| Bodily pain very mild at 50 | -.0415 (0.88) | | .2877 (4.72) | .0154 (1.21) | |
| Bodily pain mild at 50 | -.0657 (1.12) | | .4936 (6.76) | -.0053 (0.34) | |
| Bodily pain moderate at 50 | -.2265 (3.73) | | .7295 (9.46) | .0101 (0.61) | |
| Bodily pain severe at 50 | -.1580 (1.61) | | 1.0492 (8.60) | .0242 (0.95) | |
| Bodily pain very severe at 50 | -.8123 (3.93) | | 1.4199 (5.87) | -.0043 (0.08) | |
| Back pain at 42 | | | .1885 (3.12) | | |
| Life satisfaction at 50 | | .2001 (10.53) | | | |
| Life satisfaction at 46 | | .2631 (10.93) | | | |
| Life satisfaction at 42 | | .0820 (4.69) | | | |
| Male | .1078 (2.85) | .3563 (6.72) | -.6168 (12.68) | .0010 (0.10) | -.0057 (0.60) |
| Region | Yes | Yes | Yes | Yes | Yes |
| Labor force status | Yes | Yes | Yes | Yes | Yes |
| Constant | 6.8668 | 3.0729 | 1.2865 | 0.1014 | 0.1165 |
| Adjusted R$^2$ | 0.0316 | 0.1609 | 0.1301 | 0.0055 | 0.0071 |
| N | 4,121 | 5,119 | 5,180 | 4,193 | 4,847 |

Notes: personal controls are labor force status dummies in NCDS Covid at age 62.

For definition of malaise score see Table 11. It has a mean of 1.44 and the distribution is as follows

0 = 43.3%; 1 = 22.1%; 2 = 13.2%; 3 = 7.3%; 4 = 5.4%; 5 = 3.5%; 6 = 2.5%; 7 = 1.6%; 8 = 1.0% and 9 = 0.1%

Inclusion of controls for gender and pain at various ages brings the sample size down to 13,859 in column 1 of Table 6. We find that those whose mothers were single with no husband were especially unlikely to respond to the BMS, as were those whose fathers were unskilled workers at the time of the respondent's birth in 1958. Those with parents higher administrators or professionals or mangers had a lower probability. Pain at age 16 significantly raises the probability of attrition at age 44 and especially for those for whom it "certainly applies".

Column 2 of Table 6 estimates the probability of suffering chronic pain at age 44 among respondents to the BMS. The model takes the following form:

$$Pain_{44} = \beta_0 + \beta_1 pain_7 + \beta_2 pain_{11} + \beta_3 pain_{33} + \beta_4 pain_{42} + \beta_5 male + \beta_5' Xrb_0 + \beta_5' Xfb_0 \text{ (Eq 3)}$$

where chronic pain at age 44 is a function of lagged pain across the life-course, being male, region at birth and a vector capturing mother's husband's social class at birth.

Pain at sixteen is significant and positively associated with pain a quarter of a century later. Perhaps surprisingly, father's social class in the year of the respondent's birth has a statistically significant impact on reports of chronic pain 44 years later. Those whose fathers had been professionals, managers and higher-level administrators when they were born were significantly less likely to report chronic pain 44 years later. The variables predicting a higher likelihood of attrition in column 1 tend to be positively correlated with reporting chronic pain at age 44 suggesting that patterns of non-response may have downwardly biased estimates of the incidence of chronic pain in mid-life in this birth cohort. As noted earlier the BMS estimate of the incidence of chronic pain (41%) is broadly consistent with estimates of the incidence of pain in the UK reported in [38] (42%).

In the third column of Table 6 we rerun the estimates for chronic pain at age 44 but this time add the pain variable at age 11. Doing this reduces the sample size to 6,637 as the sample consists of people who were present in the PMS, NCDS2 and BMS and had non-missing values on the pain variables in childhood. Both measures of pain in childhood are positively and significantly associated with reporting chronic pain at age 44, confirming what other studies have found. Analogously, [44] shows that childhood cardiovascular risk factors (ages 3–19) impact adult cardiovascular events after a mean follow-up of 35 years.

Column 4 adds a pain variable at age 7, which is weakly significant (t = 1.7). Column 5 adds two more pain variables at ages 33 and 42 and all are significantly positive. Of note is how stable the coefficients are on the various pain variables as the variable specifications are changed.

It seems that pain is highly persistent over the life-course. The introduction of these childhood pain variables increases the coefficients on father's social class at the time of the NCDS respondent's birth.

Another way to control for attrition is to use the fitted values from column 1 to calculate an inverse probability weight which can be applied to weight the chronic pain models. For illustration purposes column 6 re-estimates the equation in column 2 using these inverse probability weights obtained for the attrition equation in column 1. Results are little changed.

## 5.2. Bodily pain at age 50 in NCDS8

**Respondents to NCDS8 were asked:** How much bodily pain have you had during the past 4 weeks (n = 9810).

The responses were as follows—none (28.5%); very mild (29.3%); mild (17.2%); moderate (17.4%); severe (6.2%) and very severe (1.6%)—so eight percent of the sample said they were in severe or very severe pain.

Pain at age fifty was highly correlated with pain at age 44. Among those reporting chronic pain at age 44 in Table 7 column 1, only 21% had reported no pain at age 50, compared with 84% of those reporting "very severe" pain at age 50.

Columns 2 and 3 of Table 7 show the distribution of pain at age 50 for those with no academic qualifications (n = 1690) versus those with a degree or higher (n = 2078) from an overall sample size of 9810. Education here is measured at NCDS8. Pain is higher among the less educated.

Table 8 uses linear estimation again to establish the correlates of pain respondents were experiencing at age 50. The dependent variable is coded from 1 = never to 6 = very severe. The structure of the equation is very similar to that in equation (3) used in Table 4.

The variable has a mean of 2.49. In column 1 we include the chronic and short pain variables from the BMS at age 44 alongside sex and region: both pain variables are positive and highly significant, though the former has a coefficient two and a half times as large as the latter. We vary controls in the next three columns, but the two coefficients remain fairly stable. In the second column we add controls for back pain at age 33 and 42 and both are significantly positive along with the chronic and short pain variables. In the third column we add controls for labor force status and education at 50. Confirming the raw data, pain is greater for those with less education, and especially high for those permanently sick or disabled. But adding controls only reduces the chronic pain coefficient from .98 to .87.

We then add father's social class at the respondent's birth from the Perinatal Mortality Study in column 4 and pain at age 16 in column 5. In what follows we exclude the pain at age 11 variable as it is always insignificant in the presence of later in life pain variables. The other coefficients are largely unchanged and indeed the coefficient on the chronic pain variable is stable to the change in specifications. The age 16 pain variable is measured before any of the respondents left school and hence is exogenous to labor market experience [45]. This group were one of the first to be subject to the Raising of the School Leaving Age (ROSLA) to sixteen that was implemented in 1972. Pain at sixteen remarkably raises the probability of pain thirty-four years later, confirming the persistence of pain over time.

## 5.3. Back pain at age 55 from NCDS9

Bodily or chronic pain are not available in NCDS9 but what is available is back pain. Respondents were asked since their prior interview if they had had recurrent back ache, a prolapsed disc, sciatica or other back problems. They were then asked if they had each of these four outcomes individually. The variable was distributed across 885 respondents as follows: any = 25.7%; with the means of the types as follows: recurrent back ache (20.1%); sciatica (10.9%); prolapsed disc (4.7%) other back ache (7.5%). 1154/8885 said one of them; 719 said two of them 303 said three and 85 said all four.

Table 9 shows the incidence (%) of having any of these back pain symptoms according to characteristics at age 55.

Back pain is highest among the less educated and the permanently sick and those who were finding it difficult to manage financially along with those in poor health.

In Table 10 we explore the determinants of backache at 55 in NCDS9 for the (0,1) outcome any backache in a similar way as we did above for bodily pain at age 50 by including a series of controls, including some from earlier sweeps of the survey.

Column 1 includes five dummy variables capturing the severity of back pain experienced at age 50 from very mild to very severe, with none the excluded category. These dummy variables are themselves jointly significant, with more severe pain at age 50 more strongly correlated with back pain at age 55.

Column 2 adds short and chronic pain at age 44, along with controls for gender, labor force status, financial situation and education at age 55. Pain a decade earlier is a significant predictor of back pain at 55. Chronic pain has more than twice as large an effect as short pain.

Column 3 adds short and long pain which are significantly positive as are the other seven pain variables. Column 4 adds pain reported at ages 7, 11 and 16 and the sample size drops markedly to 3,446. The childhood pain variables are insignificant, but their inclusion has little impact on the size or significance of the other pain variables which are all significant and positive. Of note is how similar results are in Tables 8 and 10 for bodily pain and back ache at age 50 and 55 respectively.

## 5.4. Work at age 55 in NCDS9

Whilst some studies show pain leads to job loss in the short-term, there are few studies examining the association between pain and long-term consequences for employment. We address this issue in Table 11 which presents four models estimating the probability of being in paid work, whether an employee or self-employed, at age 55 in NCDS9 as a function of chronic pain experienced at age 44 in the BMS and bodily pain at age fifty. The equation takes the following form:

$$Work_{55} = \beta_0 + \beta'_1 pain_{50} + \beta_2 cpain_{44} + \beta_3 spain_{44} + \beta_4 work_{50} + \beta_5 work_{46} + \beta_5 male + \beta'_6 Xr_{55} + \beta'_7 Xfs_{55} \quad \text{(Eq 4)}$$

where $Work_{55}$ is the paid work status of the cohort member at age 55, as a function of a vector of pain variables at age 50; region and financial situation at age 55; chronic and short-pain at age 44; lagged work status at ages 50 and 46; and being male.

Four-fifths (80.1%) of those in the survey (n = 8,973) at age 55 were in paid work. Once again, we excluded the childhood pain variables as they were insignificant.

Moderate to severe pain at age 50 lowers the probability of working at age 55, controlling for labor force status, education and a person's financial situation. The association is especially large for very severe pain. Adding controls for pain at age 44 in column 2 from the BMS shows that chronic pain is associated with a significantly lower probability of having a job. Short pain is not significant. The bodily pain at 50 coefficients are largely unchanged despite the drop in sample size by nearly 1400.

The association with severe pain at age 50 is still apparent and statistically significant in column 3 when labor force status at 50 is added, but the coefficient on chronic pain at age 44 is no longer statistically significant. This is also the case in column 4 when labor force status at age 46 is included.

We now turn from pain to examining the impact of chronic pain on subsequent measures of wellbeing years later.

## 5.5. Depression, emotional and psychiatric problems since the last interview

Respondents in NCDS9 at age 55 are asked about their mental health and whether since the last interview five years earlier they have had depression, emotional, nervous, or psychiatric problems. Where they answer 'yes' we code them '1' on a dummy variable used as the dependent variable in the OLS regression reported in Table 12. It has a mean of .152 (n = 8,875).

Table 12 examines the association between depression at age 55 and pain in earlier years. The full model in column 4 takes the following form:

$$MH_{55} = \beta_0 + \beta'_1 pain_{50} + \beta'_2 pain_{44} + \beta_3 bpain_{42} + \beta_4 bpain_{33} + \beta'_4 MH_{44} + \beta_5 male + \beta'_6 Xr_{55} + \beta'_7 X_{55} \quad \text{(Eq 5)}$$

where $MH_{55}$ is poor mental health at age 55, as a function of a vector of pain variables at ages

50 and 44; back pain at ages 42 and 32; lagged mental health at age 44; region and personal controls at age 55.

Column 1 includes personal controls, gender and the bodily pain severity at age 50. Pain at age 50 is significantly positive with larger coefficients as pain becomes more severe. In column 2 chronic pain at age 44 is significant in predicting depression at age 55 while short pain is not. The chronic pain effect remains robust to the introduction of additional lagged pain variables in column 3. These lagged pain variables at ages 42 and 33 are themselves positive and statistically significant.

As noted earlier, there is a literature indicating bi-directional associations between pain and mental health. We therefore seek to isolate the independent effect of pain at age 44 on depression at age 55 having netted out any effects associated with mental health problems at age 44. We do so by adding 8 mental health scores of various types from age 44, all of which are positive and statistically significant for mental health problems at age 55. The introduction of these variables, together with the introduction of personal controls identified in the footnote to the table, reduces the size of the chronic pain coefficient which is no longer significant in column 4. The back pain at 33 is significant (t = 2.07) that at age 42 is weakly significant (t = 1.66).

## 5.6. General Health at age 55 from NCDS9

As we noted above chronic pain tends to be higher among those in poor health. The general health variable at age 55 is coded from poor (= 1) to excellent (= 5). In the case of those with poor health at age 55 72% said they had chronic pain at 44 compared with 25% for those with excellent health. But how does pain affect future assessments of general health? We address this question in Table 13 which takes individuals' assessments of their general health at age 55 as the dependent variable. The model takes a similar form to those reported above and includes lagged assessment of health variables at age 50 in all columns and at age 46 in column 4. Conditioning on self-assessed general health at age 50, pain at 50 lowers general health at 55 without personal controls (column 1) or with them (column 2). The association increases with the severity of pain experienced at age 50. Column 3 shows that chronic but not short pain at 44 lowers general health at 55, even when conditioning on lagged general health and bodily pain at age 50. Little changes in column 4 when health status at age 46 is also included. Pain worsens general health years later.

## 5.7. Attitudes about the present and the future at age 55

If, as the literature review suggests, chronic pain can alter the way people see the world and think about their future, this may show up in models estimating attitudes at age 55. This is what we examine in Table 14 with models taking a similar form to those discussed above. We examine how people respond to three statements. The first question is "*I feel full of energy these days*". The second is "*I feel that life is full of opportunities*" while the third is "*I feel that the future looks good for me*". Each of these variables are coded 4 = often; 3 = sometimes; 2 = not often and 1 = never. For each outcome we include bodily pain at 50 as controls plus the mental health score at age 44, which is taken from BMS. The age 44 mental health score is coded 1–33. In a third of cases the value is zero and in two thirds of cases the value is three or below. Its mean is 3.4. It is used to pick up mental health issues at age 44 so as to isolate the independent effects of pain more clearly. In all six columns its coefficient is highly statistically significant and negative. The bodily pain variables are highly significant and negative. Once more we excluded the childhood pain variables as they were insignificant.

For all three variables we find chronic pain at age 44 elicits negative attitudes at age 55. In contrast short pain is statistically significant in the energy equation but not for opportunities or the views on the future. It is clear that chronic pain worsens attitudes years later.

### 5.8. Life satisfaction and Malaise scores at 46 (NCDS7) and at age 50 (NCDS8)

We now turn to examine other well-being scores in Table 15. Once more, models are structured in a similar fashion to those above. The first two sets of results examine the effects of pain at age 44 on life satisfaction at age 46 and again at age 50. The final set of estimates in the right-hand panel of Table 15 examine unhappiness using the malaise score at 50 used in [41].

The left-hand column of Table 15 estimates a single model for responses to the question asked at age 46 in NCDS7: "*how satisfied are you with the way life has turned out so far*?". This variable is scored from 0–10 (where higher scores indicate higher satisfaction) with a mean of 7.57. The variable also includes a highly significant, and positive, lagged life satisfaction variable at age 42. The chronic pain coefficient is one-and-a-half times larger than that for short pain.

The second column uses the same life satisfaction question–"*how satisfied are you with the way life has turned out so far*?" but asked at age 50 in NCDS8. Again, responses are coded 0–10. The mean falls a little to 7.28 when compared to age 46, consistent with the U-shape in age found in the literature [37]. This includes a lagged life satisfaction variable at 46 and remarkably, pain at age 44 is associated with lower life satisfaction, with chronic pain predicting lower life satisfaction a decade later.

Column 3 plugs life satisfaction at age 46 and 42 into the life satisfaction at age 50. This indicates a great deal of persistence in the life satisfaction responses; life satisfaction at age 46 has a large positive and highly significant coefficient and its introduction increases the adjusted r-squared in the model by .10. One might imagine that its introduction would knock out the effects of pain but it doesn't: chronic pain at age 44 remains negative and statistically significant, with a larger coefficient than in column 2.

The final three columns provide the mirror image on life satisfaction: they estimate unhappiness. The malaise score at NCDS8 (n = 10781), is scored from zero to 9 and has a mean of 1.488, with 44.3% of the sample scoring zero and 77.3% scoring 2 or below. It is the sum of the yes/no responses to nine variables regarding mental health at age 50 with means in parentheses:

1. feels tired most of the time (.275),

2. often feels miserable and depressed (.192),

3. often gets worried about things (.422),

4. often gets into a violent rage (.026),

5. often suddenly scared for no good reason (.084),

6. is easily upset or irritated (.254),

7. is constantly keyed up and jittery (.077),

8. whether every little thing gets on respondent's nerves (.079),

9. whether respondent's heart often races like mad (.083),

The results are striking. Regardless of model specification both chronic and short pain at age 44 impact malaise at 50 with chronic pain having the much larger effect. That is true in column 5 with a lagged life satisfaction at 46 term and in column 6 which adds a further lag life satisfaction term at age 42. Taken together Table 15 provides overwhelming evidence that chronic pain predicts happiness and raises unhappiness in subsequent years.

### 5.9. Hours of sleep at age 50 and problems falling asleep at age 44 in BMS

Table 16 examines the effects of pain at age 44 and in childhood on hours of sleep per night at age 50 and then whether the respondent was a short sleeper (reporting sleep of less than 7 hours a night) and whether the respondent had problems trying to fall asleep in the last month. Following [15] we examine short sleep and compare this to normal sleepers who get more hours a night.

For both hours of sleep and short sleep at age 50 we run two model specifications. The first two columns relate to hours of sleep and conditions on gender, region and mental health score at age 44. The second adds back pain, at 33, from NCDS5. Chronic and short pain at age 44 reduce hours of sleep six years later, with the former having bigger coefficients. The coefficients are relatively stable across the two model specifications. Pain at age 33, seventeen years earlier, has its own negative and statistically significant effect. Very similar results are found for short sleep: the probability of being a short sleeper at age 50 is increased by the experience of pain, particularly chronic pain, at age 44.

In the far-right panel of Table 16 we estimate two equations estimating correlates of having problems falling asleep at age 44. A quarter of respondents said that was the case, a good bit of the time, most of the time or all of the time (n = 8,766). It is apparent that both chronic pain and short pain affect contemporaneous problems sleeping, whereas childhood pain plays no significant role.

### 5.10. Sleep, life satisfaction, malaise and having Covid at age 62

Table 17 makes use of the NCDS Covid survey taken mostly in February of 2021. Column 1 uses hours of sleep as the dependent variable, and it looks much like the first two columns of Table 16. Bodily pain at 50 and chronic pain at 44 reduce hours of sleep. Back pain at age 55 lowers life satisfaction at age 62 with three lagged life satisfaction variables. Analogously back pain at 55 and bodily pain at 50 raise malaise scores at 62.

Finally, in the 2021 survey respondents report on whether they had had Covid. Blanchflower and Bryson [46] provide an analysis of Covid and the accompanying rise in anxiety, worry and depression in the US using data from the Census Bureaus' Household Pulse Surveys of 20020–2022. In column 4 we include back pain at 55 and bodily pain at 50 which are both insignificant, but chronic pain at age 44 is positive and statistically significant. In column 5 the chronic pain variable is significantly positive, and the short pain variable has a similar coefficient and weakly significant. Pain at age 44 predicts getting covid nearly two decades later presumably reflecting other morbidities that suggest vulnerabilities to covid.

## 6. Conclusions

In the data we use in this study, two-fifths of those in their 40s reported suffering chronic pain. A major reason for taking chronic pain seriously is that it has sizeable and persistent effects on a whole range of health and wellbeing outcomes for people, including life expectancy.

Much of the literature examining the '*effects*' of pain is either cross-sectional, or else examines short-term effects of pain using panel data tracking individuals over some months or, occasionally, a few years. Often effects are estimated for non-random sub-populations such as those who have sought treatment for pain. There are exceptions but they are rare. Our paper takes a very different approach. We take a cohort of all those born in a single week in March in 1958 in Britain track the consequences of short pain and chronic pain in mid-life at age 44 on health, wellbeing and labor market outcomes decades later in life (at ages 50–62). We also show the impact of backache and bodily pain at other ages. We examine a very broad range of outcomes in mid-life including pain, general health, depression, malaise, life satisfaction,

energy levels, optimism, paid work, and sleep quality as well as catching covid. Ours is a *prospective* study, thus avoiding the problems of potential recall biases.

We find those suffering both short-term and chronic pain at age 44 continue to report pain and poor general health later in life. However, the associations are much stronger for those with chronic pain. Chronic pain at age 44 is also associated with a range of poor mental health outcomes at ages 50–62 including depression, pessimism about the future, malaise and lower life satisfaction, as well as poor general health, poor sleep, short sleep and joblessness.

Of particular note is our finding that pain at age 44 predicts whether a respondent had Covid at age 62. This suggests that pain is picking up broader vulnerabilities that make people susceptible to Covid.

We also find that social class of the respondent's mother's husband–not necessarily their father—in 1958 reported at the time of the respondent's birth matters for a whole host of outcomes, including pain in later life. Those with fathers in less skilled occupations–such as semi-skilled and unskilled occupations–are less likely to respond to questions on pain years later. Those with a father with a professional, or higher administrator were more likely to respond. Given that we show that pain rises according to occupational and educational status it seems likely that the estimates we present likely *understate* pain years later. So, any bias in the data due to attrition likely works against finding evidence of the transmissibility of pain through life as those with higher levels of pain are more likely to attrit. Of note though is how stable our estimates are to the inclusion of different variables from various sweeps of the NCDS with somewhat different response rates and hence universes.

Whereas, as *The Lancet* article indicates, only now are people in the UK asking for chronic pain to be taken 'more seriously', the debate about pain and what to about it has been raging for some time in the United States. Some argue there is a pain pandemic in the United States, one which is being treated unsuccessfully through opioids [47, 48]. According to the Centers for Disease Control and Prevention drug overdose deaths in the United States in 2021 increased half as much as in 2020 but are still up 15%. Provisional data from CDC's National Center for Health Statistics indicate there were an estimated 107,622 drug overdose deaths in 2021. Overdose deaths involving opioids increased from an estimated 70,029 in 2020 to 80,816 in 2021. Overdose deaths from synthetic opioids (primarily fentanyl), psychostimulants such as methamphetamine, and cocaine also continued to increase in 2021 compared to 2020 [49].

Despite an explosion in opioid prescribing in the United States, pain has not subsided [42]. One reason for this appears to be that opioids are relatively ineffective in tackling chronic pain. Recent studies have shown that other drugs are more effective than opioids in treating pain. For instance, Moore and Hersh [50, p. 898] concluded that 325 milligrams of acetaminophen taken with 200 milligrams of ibuprofen provides better pain relief than oral opioids following wisdom tooth extraction.

Blicher and Pryles [51] come to a similar conclusion with regards to managing dental pain in general: "*800mg of ibuprofen is demonstrably more effective in managing severe dental pain than other available prescription analgesics, including narcotic compounds. Furthermore, the combination of ibuprofen (Advil) and acetaminophen (Tylenol) offers greater pain relief than either medication alone and significantly more than the combination of acetaminophen and opioid medication both following endodontic treatment and third molar extraction*" (51, p.56). Similarly, the review by Lewis et al. [52] finds that non-opioid medications provided some positive global effect in treating sciatica while opioids did not.

We do not observe medication taken in our data, for the UK so our estimates of pain's effect on subsequent health, wellbeing and labor market outcomes is net of such treatments. However, our evidence on the persistence of pain across the life-course suggests efforts to counter it have not been wholly successful. We have shown that it is, in part, passed from one generation

to the next, with those from lower social classes suffering most. Pain appears to be another source of inter-generational disadvantage, and one that is potentially as problematic as other aspects of social deprivation.

## Supporting information

**S1 Appendix. Pain questions from 2002–2004 biomedical study.**
(DOCX)

## Acknowledgments

We thank the ESRC Data Archive for access to the National Child Development Study data.

## Author Contributions

**Conceptualization:** David G. Blanchflower, Alex Bryson.

**Formal analysis:** David G. Blanchflower, Alex Bryson.

**Investigation:** Alex Bryson.

**Methodology:** David G. Blanchflower, Alex Bryson.

**Writing – original draft:** David G. Blanchflower, Alex Bryson.

**Writing – review & editing:** David G. Blanchflower, Alex Bryson.

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
