## [Decision Letter · Decision Letter 0]

12 May 2022

PONE-D-21-29758The Consequences of Chronic Pain in Mid-Life Evidence from the National Child Development SurveyPLOS ONE

Dear Dr. Bryson,

Thank you for submitting your manuscript to PLOS ONE. After careful consideration, we feel that it has merit but does not fully meet PLOS ONE’s publication criteria as it currently stands. Therefore, we invite you to submit a revised version of the manuscript that addresses the points raised during the review process.

The manuscript has been evaluated by two reviewers, and their comments are available below.

You may see that reviewer 2 has made references related to the novelty of the work. Please note that PLOS ONE does not judge mansucript based on perceived novelty, and we have not taken this into consideration during the decision-making progress. However, the reviewer has also recommended discussions in the Introduction explaining how the current study further contributes to scientific understanding. We encourage that attention is paid to this aspect of the reviewer's comments during revision.

Furthermore reviewer 1 has also identified potential causal language used throughout the conclusions as this is s cross sectional study we do not believe that the study design can infer causation. Please note that our publication criteria 4 requires that the conclusions of a manuscript are presented in an appropriate fashion and are supported by the data (http://journals.plos.org/plosone/s/criteria-for-publication#loc-4).

Could you please revise the manuscript to carefully address the concerns raised?

We look forward to receiving your revised manuscript.

Kind regards,

Lucinda Shen, MSc

Staff Editor

PLOS ONE

**Journal requirements:**

“Alex Bryson thanks the Health Foundation for funding (grant number 789112).”

“Alex Bryson thanks the Health Foundation for funding (grant number 789112). We thank the ESRC Data Archive for access to the National Child Development Survey data.”

“Alex Bryson thanks the Health Foundation for funding (grant number 789112).”

“there are no competing interests”

6. Please note that in order to use the direct billing option the corresponding author must be affiliated with the chosen institute. Please either amend your manuscript to change the affiliation or corresponding author, or email us at plosone@plos.org with a request to remove this option.

Reviewers' comments:

Reviewer's Responses to Questions

**Comments to the Author**

1. Is the manuscript technically sound, and do the data support the conclusions?

Reviewer #1: Yes

Reviewer #2: Partly

2. Has the statistical analysis been performed appropriately and rigorously? 

Reviewer #1: Yes

Reviewer #2: Yes

3. Have the authors made all data underlying the findings in their manuscript fully available?

Reviewer #1: Yes

Reviewer #2: Yes

4. Is the manuscript presented in an intelligible fashion and written in standard English?

Reviewer #1: Yes

Reviewer #2: Yes

5. Review Comments to the Author

Reviewer #1: This manuscript deals with the consequences of short and chronic pain in mid-life using data from all those born in a single week in Britain in 1958.

Comments:

-Please make the codes/programs publicly available with a readme file, and provide a link in the manuscript.

-Typos: I found just a few typos throughout. See, for instance, p.1 to suffer

-Please remove all causal claims throughout. For instance, p.1 "The partial associations between chronic pain and subsequent health and labor market outcomes are credibly causal since they persist even in the presence of lagged pain measured earlier in life". To me this is not causal, but fascinating correlations.

-Pain is self-reported, so be careful with sentences such as "with pain incidence being highest among those not in the labor force".

-For attrition, discuss magnitude for key variables and not only statistical significance. For instance, "We find that those whose mothers were single with no husband were especially unlikely to respond to the BMS, as were those whose fathers were unskilled workers at the time of their birth." Is the effect large in magnitude? This will help the readers better understand if attrition is a serious concern.

-Could you provide a table similar to table 4 (in the appendix or extra columns in Table 4) where the sample is balanced for all columns?

-Sometimes you write 50, sometimes fifty. Similarly for 44 and other numbers, Why? I am not a native speaker, but it seems to me that only numbers below 10 should be written in letters. That's obviously a detail and there might be a good reason for that.

-Show or mention that the results are robust to using ordered probit (e.g., Table 5) or probit/logit (e.g., Table 6).

-Do not drop statistically insignificant results from the tables (e.g., keep Pain at age 16 in Table 7).

-Interpret the point estimates instead or in addition to reporting the coefficient (e.g., -0.03, p.11).

-"This paper supports the conclusion in a recent Lancet editorial that “chronic pain is real [and] deserves to be taken more seriously” (Lancet, 2021)." I do not see how your paper supports this statement. More generally, do not overstate your findings in the conclusion and focus on what your paper informs us rather than vague statements about how big of a deal chronic pain is.

Reviewer #2: Referee report of "The Consequences of Chronic Pain in Mid-Life Evidence from the National Child Development Survey”

Summary:

Using the National Child Development Survey, this paper examines: (i) the persistence of pain over the life cycle and (ii) how pain in adulthood predicts health, wellbeing and labor market outcomes later in life.

General comments:

This paper studies an important topic. Pain has become a worldwide issue and deserves to be taken seriously. However, I have some major reservations with the paper in its current form.

1) Contribution

The first key point of the paper is to document the persistence of pain over the life cycle. Specifically, the authors estimate the relationship between pain experienced in childhood at age 16 and pain in adulthood measured at ages 44, 50 and 55. This is an important point, but not novel. As stated by the authors, previous studies have already emphasized the association between pain in childhood and subsequent pain in adulthood.

The second key point is to show that pain in adulthood affects a wide range of health, wellbeing and labor force participation outcomes 5 to 10 years later. This component of the paper is more novel than the first, but again, it is not new. There are some studies which document the relationships between pain and subsequent health, wellbeing and labor market participation outcomes. This begs the question, what is the contribution from this work? Most previous studies rely on cross-sectional variations or panel data which only examine short-term effects of pain over a few years. Documenting that the results extend over a decade is valuable, but the main substantive point (that pain is related to lower health, unemployment and unhappiness) has already been made in existing work. I strongly suggest the authors to spell out the advantages of the cohort data in more detail and what new do we really learn from their analysis that we did not know before? While I agree cohort data can be helpful to investigate long-term relationships, I think that the same analysis could be done using panel data over 10 years and would have the advantage to control for unobserved individual time-invariant differences through person fixed effects. It would also allow to test for adaptation, which is not the case here.

2) Presentation

Overall, I find the paper not very well written. The authors need to clearly state (i) what are the main questions they are addressing in the paper, (ii) at which ages the different variables (pain, health, labor market outcomes, etc.) are measured, and (iii) what are the equations they are estimating. To do so, I would recommend the authors to include a table or a timeline which depicts the different variables available at the different ages. I would also include a method section, which would clearly describe the different equations estimated in the tables. Throughout the paper, I was always confused about exactly what relationships we are looking at and at which ages.

Related to the previous point, I would recommend to use always the same set of variables and controls in the regressions when looking at the long-term effects of pain on health, wellbeing, paid work, and sleep. Even if some variables become not significant, it would help for consistency. I would also suggest to present the results in a consistent way: either depicting all coefficients or presenting only a subset of them (with full tables in an appendix for instance).

3) Data & attrition

The authors provide some information about the NCDS data and the variables used. However, I would have liked to see a table with some descriptive statistics of the main variables used in the analysis for the estimated sample.

As the authors acknowledge, attrition could be an issue here if those who remain in the sample are individuals which are less likely to suffer from chronic pain. This would downwardly bias the estimates. However, the way these attrition tests are introduced is a bit odd. We don’t have any explanation of why the authors are doing these tests. Moreover, there is no attempt to deal with this problem in the main analysis. For instance, the authors could have introduced inverse probability weighting to give more weights to individuals more likely to attrit from the sample.

4) Empirical strategy

The authors argue in the introduction that “the partial associations between chronic pain and subsequent health and labor market outcomes are credibly causal since they persist even in the presence of lagged pain measured earlier in life and having controlled for parental and familial background in childhood, as well as a wide range of physical and mental health ailments reported in mid-life”. Although I agree that cohort data allow to somewhat deal with reverse causality, we could still argue that omitted variables and self-reporting bias are likely to affect the results. For instance, as most of the variables are self-reported we may expect some spurious correlations due to this. Although as pain is largely persistent over the life cycle, it could well be the case that unhappy people are more likely to suffer from pain, which in turn increases unhappiness. We would not be able to disentangle the cause from the effect using these cohort data. Therefore, I would suggest the authors to be more careful in their analysis, and acknowledge that they only capture correlations here.

5) Variable choices

The authors investigate the relationship between father’s social class in childhood and pain in adulthood. The paper should discuss in more detail why the authors chose to focus on this relationship and why not including other family characteristics, like father’s and mother’s health outcomes, employment status, income, marital status, etc.

6. PLOS authors have the option to publish the peer review history of their article (what does this mean?). If published, this will include your full peer review and any attached files.

Reviewer #1: No

Reviewer #2: No

---

## [Author Response · Author response to Decision Letter 0]

28 Jun 2022

3rd June 2022

Dear Editor,

PONE-D-21-29758: Chronic Pain: Evidence from the National Child Development Survey

Thank you for giving us the opportunity to revise the paper. We have addressed all the points raised by you and the two reviewers. We provide a summary of how we have done so below, replicating the reviewer comments followed by our responses denoted with ‘Answer’.

We look forward to your thoughts and are very happy to provide further clarifications or undertake any further revisions as you deem necessary.

As regards funding disclosure, Alex Bryson thanks the Health Foundation for funding (grant number 789112). The funders had no role in study design, data collection and analysis, decision to publish, or preparation of the manuscript.

Many thanks.

Alex Bryson

Points Raised by the Editor

1. You may see that reviewer 2 has made references related to the novelty of the work. Please note that PLOS ONE does not judge manuscript based on perceived novelty, and we have not taken this into consideration during the decision-making progress. However, the reviewer has also recommended discussions in the Introduction explaining how the current study further contributes to scientific understanding. We encourage that attention is paid to this aspect of the reviewer's comments during revision.

1. Answer – There are a number of ways in which the paper is novel. In particular, it is the only paper we are aware of that tracks individuals’ experience of pain over the whole life-course from birth to age 62. It is also the only paper we are aware of that examines the association between pain at various ages and a range of subsequent labour market and wellbeing outcomes. The new draft also incorporates new results for the COVID period, including the probability of getting COVID, which nobody will have seen before. Our contribution in relation to scientific understanding is that we show just how persistent pain is over the life-course, and how pain experienced even many years previously can affect subsequent health, wellbeing and labour market prospects. We have noted these points at various junctures in the redraft.

2. Furthermore reviewer 1 has also identified potential causal language used throughout the conclusions as this is a cross sectional study we do not believe that the study design can infer causation. 

2. Answer. We do look at the GWP file which is cross-sectional. The majority of the paper uses a birth cohort data file, the NCDS which is not a cross-sectional study. We follow the same 18,558 people over 62 years of their lives in twelve waves of the data file. This is as far from a cross-sectional study as any data file that exists in the world. We have noted that pain in early life has strong predictive power for pain later in life. We agree there are lots of things that pain is proxying for (general health, sensitivity, mental health etc.) so we have taken out all the causal claims.

3. Please note that our publication criteria 4 requires that the conclusions of a manuscript are presented in an appropriate fashion and are supported by the data (http://journals.plos.org/plosone/s/criteria-for-publication#loc-4).

3. Answer. We have done that

Reviewer #1: 

This manuscript deals with the consequences of short and chronic pain in mid-life using data from all those born in a single week in Britain in 1958.

Answer. Actually it also deals with the consequences of having pain at ages 11, reported in 1969 and age 16, reported in 1984 on outcomes decades later. We have removed ‘mid-life’ from the title accordingly. We have now extended the data through 2021 and have added a new sweep of the data file.

Comments:

1. -Please make the codes/programs publicly available with a readme file and provide a link in the manuscript.

1. Answer. We are providing a readme file.

2. Typos: I found just a few typos throughout. See, for instance, p.1 to suffer

2 Answer. Thank you. We have eradicated these errors.

3-Please remove all causal claims throughout. 

3 Answer. We have done that as suggested and we have changed the title in light of this.

4. For instance, p.1 "The partial associations between chronic pain and subsequent health and labor market outcomes are credibly causal since they persist even in the presence of lagged pain measured earlier in life". To me this is not causal, but fascinating correlations.

4. Answer. We now say that pain in early life has strong predictive power for pain later in life and dropped causal claims.

5. Pain is self-reported, so be careful with sentences such as "with pain incidence being highest among those not in the labor force".

5 Answer. We now say self-reported pain is highest among those out of the labour force from Tables 1 and 2. 

6. For attrition, discuss magnitude for key variables and not only statistical significance. For instance, "We find that those whose mothers were single with no husband were especially unlikely to respond to the BMS, as were those whose fathers were unskilled workers at the time of their birth." Is the effect large in magnitude? This will help the readers better understand if attrition is a serious concern.

6 Answer. We have now addressed the magnitude of the attrition effects and reported characteristics. We suspected that attrition would mean that our pain coefficients are likely downward biased with regards to subsequent pain because those most in pain are most likely to attrit. In fact, when we rerun estimates with the inverse of the probability of attrition in column 6 of Table 4 results do not differ much. And we now report rates to get a sense of magnitude.

7. Could you provide a table similar to table 4 (in the appendix or extra columns in Table 4) where the sample is balanced for all columns?

7.Answer. Done with an extra column in Table 4.

8. Sometimes you write 50, sometimes fifty. Similarly for 44 and other numbers, Why? I am not a native speaker, but it seems to me that only numbers below 10 should be written in letters. That's obviously a detail and there might be a good reason for that.

8. Answer. We haven’t heard of this before. We have tried to be consistent.

9. -Show or mention that the results are robust to using ordered probit (e.g., Table 5) or probit/logit (e.g., Table 6

9. Answer We have amended the text accordingly.

10. Do not drop statistically insignificant results from the tables (e.g., keep Pain at age 16 in Table 7).

10. Answer. We prefer to retain our current approach which is to test for the role played by pain across the life-course. Where statistical tests indicate lagged pain plays no role we exclude that variable. This is important, not least in some cases retention of such lagged pain variables can reduce sample sizes quite a bit. Instead we simply report what happens when they are included. There is a trade off with sample sizes.

11. -Interpret the point estimates instead or in addition to reporting the coefficient (e.g., -0.03, p.11).

11. We have incorporated discussion of effect size.

12. -"This paper supports the conclusion in a recent Lancet editorial that “chronic pain is real [and] deserves to be taken more seriously” (Lancet, 2021)." I do not see how your paper supports this statement. More generally, do not overstate your findings in the conclusion and focus on what your paper informs us rather than vague statements about how big of a deal chronic pain is.

12. Answer We have removed this comment.

Reviewer #2: Referee report of "The Consequences of Chronic Pain in Mid-Life Evidence from the National Child Development Survey”

Summary:

Using the National Child Development Survey, this paper examines: (i) the persistence of pain over the life cycle and (ii) how pain in adulthood predicts health, wellbeing and labor market outcomes later in life.

General comments:

This paper studies an important topic. Pain has become a worldwide issue and deserves to be taken seriously. 

Answer. Thank you.

However, I have some major reservations with the paper in its current form.

1. Contribution

The first key point of the paper is to document the persistence of pain over the life cycle. Specifically, the authors estimate the relationship between pain experienced in childhood at age 16 and pain in adulthood measured at ages 44, 50 and 55. This is an important point, but not novel. As stated by the authors, previous studies have already emphasized the association between pain in childhood and subsequent pain in adulthood.

1. Answer. Ours is a more comprehensive study than has been done previously. Its unique design means we can control for characteristics at birth as well as pre labour market entry. We have multiple lags of pain over the life course including chronic, long lasting, pain. These are often introduced together to work out which survive statistically. We know of no other papers that do this. We show pain at age 44 impacts life satisfaction, having energy and being optimistic, general health, sleep, depression and the probability of work, as well as experiencing back pain at 50 and 55. The birth cohort design means we can use data obtained directly years earlier and not by recall including while at school and from parents and teachers. These results are novel. The new analysis of COVID and pain is also novel. We have explained our contribution in the revised introduction.

2. The second key point is to show that pain in adulthood affects a wide range of health, wellbeing and labor force participation outcomes 5 to 10 years later. This component of the paper is more novel than the first, but again, it is not new. There are some studies which document the relationships between pain and subsequent health, wellbeing and labor market participation outcomes. 

2. Answer – Thank you for this comment. We are aware of these papers and have reviewed them in Section 2. Our Section 2.1 discusses those linking pain to subsequent health; section 2.2 reviews those linking pain to subsequent labour market participation; and section 2.3 reviews those linking pain to subsequent wellbeing. But for the reasons noted above, our paper contributes over and above that literature. If there are specific papers we have missed in our review we’d be keen to incorporate them.

3. This begs the question, what is the contribution from this work? Most previous studies rely on cross-sectional variations or panel data which only examine short-term effects of pain over a few years. Documenting that the results extend over a decade is valuable, but the main substantive point (that pain is related to lower health, unemployment and unhappiness) has already been made in existing work. 

3. Answer. The contribution of our study is to show effects that extend over a period of fifty years. Plus, we put all the evidence of the effects of pain on multiple variables and the results are essentially the same. We show (Table 4) that that abdominal pain at 11 and 16 impacts pain at age 44, controlling for pain at 42 and at 33. We show further (Table 5) that pain at 16 impacts pain at age 50, three decades later, controlling for pain at 50, 42 and at 33. We show (Table 6) that pain at 55 is impacted by chronic pain at 44, controlling for pain at 50, 42 and 33. This has not been shown before. We also show that chronic pain at 44 impacts depression and other attitudes at 55, life satisfaction at 46 and 50 and the malaise score. We show that pain also predicts Covid, which hasn't been shown before

4. I strongly suggest the authors to spell out the advantages of the cohort data in more detail and what new do we really learn from their analysis that we did not know before? While I agree cohort data can be helpful to investigate long-term relationships, I think that the same analysis could be done using panel data over 10 years and would have the advantage to control for unobserved individual time-invariant differences through person fixed effects. It would also allow to test for adaptation, which is not the case here.

4. Answer. This is for a group of people of the same age while a panel study such has GSOEP or BHPS includes those of varying age. We do look at adaptation as we have data at age sixteen so that is wrong. We use lagged personal dependent variables which is effectively conditioning away personal fixed components. We show pain at 44 predicts who got Covid twenty years later.

5.) Overall, I find the paper not very well written. The authors need to clearly state (i) what are the main questions they are addressing in the paper, (ii) at which ages the different variables (pain, health, labor market outcomes, etc.) are measured, and (iii) what are the equations they are estimating. 

5) Done

6) To do so, I would recommend the authors to include a table or a timeline which depicts the different variables available at the different ages. I would also include a method section, which would clearly describe the different equations estimated in the tables. Throughout the paper, I was always confused about exactly what relationships we are looking at and at which ages.

6) Thank you for this suggestion. We have added the timeline.

7) Related to the previous point, I would recommend to use always the same set of variables and controls in the regressions when looking at the long-term effects of pain on health, wellbeing, paid work, and sleep. Even if some variables become not significant, it would help for consistency. I would also suggest to present the results in a consistent way: either depicting all coefficients or presenting only a subset of them (with full tables in an appendix for instance).

7. Answer - We have done this in Table 2 using the GWP data as suggested. However, there is a trade off with sample sizes using the NCDS if we include all lags in all models due to some respondents appearing in only a subset of waves. We therefore prefer to retain our current approach which is to test for the role played by pain across the life-course. Where statistical tests indicate lagged pain plays no significant role, we exclude that variable. This is important, not least in some cases retention of such lagged pain variables can reduce sample sizes quite a bit. Instead, we simply report what happens when they are included. We are struck by how little the sample estimates change in the face of specification changes in variables and varying sample sizes as we use variables from various sweeps of the NCDS.

8. The authors provide some information about the NCDS data and the variables used. However, I would have liked to see a table with some descriptive statistics of the main variables used in the analysis for the estimated sample.

8. We have added this.

9. As the authors acknowledge, attrition could be an issue here if those who remain in the sample are individuals which are less likely to suffer from chronic pain. This would downwardly bias the estimates. However, the way these attrition tests are introduced is a bit odd. We don’t have any explanation of why the authors are doing these tests. Moreover, there is no attempt to deal with this problem in the main analysis. For instance, the authors could have introduced inverse probability weighting to give more weights to individuals more likely to attrit from the sample.

9. We now report some estimates in Table 4 using inverse probability weights and the results are essentially the same.

10. The authors argue in the introduction that “the partial associations between chronic pain and subsequent health and labor market outcomes are credibly causal since they persist even in the presence of lagged pain measured earlier in life and having controlled for parental and familial background in childhood, as well as a wide range of physical and mental health ailments reported in mid-life”. Although I agree that cohort data allow to somewhat deal with reverse causality, we could still argue that omitted variables and self-reporting bias are likely to affect the results. For instance, as most of the variables are self-reported we may expect some spurious correlations due to this. Although as pain is largely persistent over the life cycle, it could well be the case that unhappy people are more likely to suffer from pain, which in turn increases unhappiness. We would not be able to disentangle the cause from the effect using these cohort data. Therefore, I would suggest the authors to be more careful in their analysis and acknowledge that they only capture correlations here.

10. Answer. We have removed the causal language as suggested.

11. The authors investigate the relationship between father’s social class in childhood and pain in adulthood. The paper should discuss in more detail why the authors chose to focus on this relationship and why not including other family characteristics, like father’s and mother’s health outcomes, employment status, income, marital status, etc.

11. Answer Father’s and mother’s health outcomes, employment status, income, marital status are all variables and we have several subsequent observations on these, frequently with lots of missing values. The benefit of using data from the original Perinatal Mortality Study is we have high numbers of observations (n=17000/18558) including on the stillbirths and neonatal deaths. Many other studies find that social class at birth is a crucial predictor of life course outcomes for social mobility.

We actually use data at the time of the child's birth, reported by the mother. These are the variables we had available to us and it would be impossible to go back to 1958 and ask different questions. Unlike in other panel data files mother's health was reported on by their doctor in 1958 at the time of the child's birth. So, we focus on data at birth, which is clearly novel. The benefit of these reports is there is only a little attrition, as some children born in the birth week are added later but we add a control for missing which is always insignificant.

Of course, mother's and father's health outcomes can vary over time. Every other panel data file presumably doesn’t have this as the respondent reports it. We know of no panel data files that have comparable details at birth for survivors and non-survivors.

---

## [Decision Letter · Decision Letter 1]

18 Jul 2022

PONE-D-21-29758R1Chronic Pain: Evidence from the National Child Development SurveyPLOS ONE

Dear Dr. Bryson,

Thank you for submitting your manuscript to PLOS ONE. After careful consideration, we feel that it has merit but does not fully meet PLOS ONE’s publication criteria as it currently stands. Therefore, we invite you to submit a revised version of the manuscript that addresses the points raised during the review process.

The manuscript has been re-evaluated by two reviewers, and their comments are available below.

The reviewers have raised a few concerns that need attention. They request additional information on methodological aspects such as the inclusion of equations and revisions to the study reporting.

Could you please revise the manuscript to carefully address the concerns raised?

<o:p></o:p>

We look forward to receiving your revised manuscript.

Kind regards,

Johannes Stortz

Staff Editor

PLOS ONE

Journal Requirements:

Reviewers' comments:

Reviewer's Responses to Questions

**Comments to the Author**

1. If the authors have adequately addressed your comments raised in a previous round of review and you feel that this manuscript is now acceptable for publication, you may indicate that here to bypass the “Comments to the Author” section, enter your conflict of interest statement in the “Confidential to Editor” section, and submit your "Accept" recommendation.

Reviewer #1: All comments have been addressed

Reviewer #2: (No Response)

2. Is the manuscript technically sound, and do the data support the conclusions?

Reviewer #1: Yes

Reviewer #2: Yes

3. Has the statistical analysis been performed appropriately and rigorously? 

Reviewer #1: Yes

Reviewer #2: Yes

4. Have the authors made all data underlying the findings in their manuscript fully available?

Reviewer #1: Yes

Reviewer #2: Yes

5. Is the manuscript presented in an intelligible fashion and written in standard English?

Reviewer #1: Yes

Reviewer #2: No

6. Review Comments to the Author

Reviewer #1: (No Response)

Reviewer #2: I would like to thank the authors for their revisions and for having taken into account most of my comments. I still have some remarks, which are listed below:

- At the end of the literature review section, I miss a paragraph clearly stating the contribution of this paper with respect to previous studies

- I would also have liked to see some equations. It will help the readers to understand: what controls are included in the regressions (lags, personal characteristics, region, etc.) and what is the empirical strategy.

- Sometimes you write column two, sometimes column 2. This is a minor comment but I would suggest to be consistent.

7. PLOS authors have the option to publish the peer review history of their article (what does this mean?). If published, this will include your full peer review and any attached files.

Reviewer #1: No

Reviewer #2: No

---

## [Author Response · Author response to Decision Letter 1]

25 Jul 2022

25th July 2022

PONE-D-21-29758R1: Chronic Pain: Evidence from the National Child Development Survey

Dear Dr. Stortz,

Thank you for giving us the opportunity to revise our paper further. Referee 1 was content with the paper. So our revisions are a response to the three comments made by Referee 2. We replicate the reviewer’s comments followed by a response to them below.

Reviewer #2: I would like to thank the authors for their revisions and for having taken into account most of my comments. I still have some remarks, which are listed below:

1. Reviewer Comment: At the end of the literature review section, I miss a paragraph clearly stating the contribution of this paper with respect to previous studies

Our response: Thank you for this suggestion. We have added a paragraph accordingly at the end of the literature section which reads as follows:

Most longitudinal investigations of links between pain and subsequent health and labor market outcomes focus on short-term change captured over the space of a year, sometimes two, and are thus unable to shed any light on the longer-term effects of pain. We contribute to the literature with data permitting us to examine longer-term relationships over decades, beginning with experience of pain in childhood, to assess the effects of pain on a wider range of health, wellbeing and labor force outcomes than is ordinarily available in a single data set. We show that, for a cohort born in Britain in a single week in 1958 those reporting aches and pains at age 44 report poorer health, wellbeing and labor market outcomes over the following two decades through age 62. The partial associations between chronic pain and subsequent health and labor market outcomes persist even in the presence of lagged pain measured earlier in life and having controlled for parental and familial background in childhood, as well as a wide range of physical and mental health ailments reported in mid-life. We also find that chronic pain at age 44 in 2002 predicts having Covid nearly two decades later in 2021. 

2. Reviewer Comment: I would also have liked to see some equations. It will help the readers to understand: what controls are included in the regressions (lags, personal characteristics, region, etc.) and what is the empirical strategy.

Thank you for this suggestion. The paper now incorporates five equations relating to tables 2-8. We have not replicated them for Tables 9 onwards because they take a very similar form so it gets repetitive.

3. Reviewer comment 3: Sometimes you write column two, sometimes column 2. This is a minor comment but I would suggest to be consistent.

Thank you. We have altered the text accordingly.

We hope the revision deals with the points made and but if you need further revisions we will undertake them quickly.

Many thanks.

---

## [Decision Letter · Decision Letter 2]

12 Sep 2022

Chronic Pain: Evidence from the National Child Development Survey

PONE-D-21-29758R2

Dear Dr. Bryson,

We’re pleased to inform you that your manuscript has been judged scientifically suitable for publication and will be formally accepted for publication once it meets all outstanding technical requirements.

Sincerely yours,

Yann Benetreau, PhD

Division Editor, PLOS ONE

Additional Editor Comments (optional):

Please note the final comments from reviewer 2, which you may take into account for the final version of your manuscript.

Reviewers' comments:

Reviewer's Responses to Questions

**Comments to the Author**

1. If the authors have adequately addressed your comments raised in a previous round of review and you feel that this manuscript is now acceptable for publication, you may indicate that here to bypass the “Comments to the Author” section, enter your conflict of interest statement in the “Confidential to Editor” section, and submit your "Accept" recommendation.

Reviewer #2: All comments have been addressed

2. Is the manuscript technically sound, and do the data support the conclusions?

Reviewer #2: Yes

3. Has the statistical analysis been performed appropriately and rigorously? 

Reviewer #2: Yes

4. Have the authors made all data underlying the findings in their manuscript fully available?

Reviewer #2: Yes

5. Is the manuscript presented in an intelligible fashion and written in standard English?

Reviewer #2: Yes

6. Review Comments to the Author

Reviewer #2: Thanks for taking into account my remarks. A last one: in equation 1, you decided to indicate that the dependent variable is defined at the individual level, with small i. I would then indicate that age, education, etc... are also defined at the individual level, adding a small i to these variables, or I would delete it from P (to be consistent with your other equations). You could also have included error terms in all your equations.

7. PLOS authors have the option to publish the peer review history of their article (what does this mean?). If published, this will include your full peer review and any attached files.

Reviewer #2: No

---

## [Editor Report · Acceptance letter]

28 Sep 2022

PONE-D-21-29758R2 

Chronic Pain: Evidence from the National Child Development Study 

Dear Dr. Bryson:

I'm pleased to inform you that your manuscript has been deemed suitable for publication in PLOS ONE. Congratulations! Your manuscript is now with our production department. 

Kind regards, 

on behalf of

Dr. Yann Benetreau 

Staff Editor

PLOS ONE